



# Optimal Estimation of Water Vapour Profiles using a Combination of Raman Lidar and Microwave Radiometer

Andreas Foth[1] and Bernhard Pospichal[1,2]

[1]Leipzig Institute for Meteorology, University of Leipzig, Leipzig, Germany
[2]Institute for Geophysics and Meteorology, University of Cologne, Cologne, Germany

*Correspondence to:* Andreas Foth (andreas.foth@uni-leipzig.de)

**Abstract.** In this work, a two-step algorithm to obtain water vapour profiles from a combination of Raman lidar and microwave radiometer is presented. Both instruments were applied during an intensive two-month measurement campaign (HOPE) close to Jülich, western Germany, during spring 2013. To retrieve reliable water vapour information from inside or above the cloud a two-step algorithm is applied. The first step is a Kalman filter that extends the profiles, truncated at cloud base, to the full height range (up to 10 km) by combining previous information and current measurement. Then the complete water vapour profile serves as input to the one-dimensional variational (1D-VAR) method, also known as optimal estimation. A forward model simulates the brightness temperatures which would be observed by the microwave radiometer for the given atmospheric state. The profile is iteratively modified according to its error bars until the modelled and the actually measured brightness temperatures sufficiently agree. The functionality of the retrieval is presented in detail by means of case studies under different conditions. A statistical analysis shows that the availability of Raman lidar data (night) improves the accuracy of the profiles even under cloudy conditions. During the day, the absence of lidar data results in larger differences in comparison to reference radiosondes. The data availability of the full height water vapour lidar profiles of 17 % during the two-month campaign is significantly enhanced to 60 % by applying the retrieval. The bias with respect to radiosonde and the retrieved a posteriori uncertainty of the retrieved profiles clearly show that the application of the Kalman filter considerably improves the accuracy and quality of the retrieved mixing ratio profiles.

## 1 Introduction

In accordance with the latest report of the Intergovernmental Panel on Climate Change (IPCC), water vapour plays a key role in the description of the thermodynamic state of the atmosphere (Hartmann et al., 2013) and it is the most important greenhouse gas (Twomey, 1991). Its amount in the atmosphere is controlled mostly by the air temperature, rather than by emissions. Therefore, tropospheric water vapour is considered as a feedback agent more than a forcing to climate change (Soden and Held, 2006). The water vapour amount is highly variable in space and time, since it can considerably increase due to evaporation or decrease due to condensation and precipitation (Stevens and Bony, 2013). Furthermore, the latent heat strongly influences the energy cycle. The typical residence time of water vapour in the atmosphere amounts to ten days (Myhre et al., 2013). Due to its



spatio-temporal variability and its involvement in many atmospheric processes (e.g. cloud formation) it is difficult to properly implement water vapour in climate models (Held and Soden, 2000; Tompkins, 2002).

In the last decades, the resolution of atmospheric circulation models has been improved, more atmospheric processes have been incorporated and the parametrisations of physical processes have been improved (Randall et al., 2007). In order to eval-
uate and improve model forecasts, parametrisation schemes and satellite retrievals, the observations need to be enhanced. Uncertainties in both observations and modelling of water vapour strongly affect the representation of clouds and precipitation in climate models and predictions. For that reason the German research project High Definition Clouds and Precipitation for advancing Climate Prediction (HD(CP)$^2$) was initiated aiming to improve the clouds and precipitation representation in models and to quantify the errors associated. One part within the HD(CP)$^2$ initiative was the intensive observation campaign HD(CP)$^2$
Observational Prototype Experiment (HOPE) in Jülich (Macke et al., 2016). Data from this campaign will be used in this work which presents a retrieval of water vapour profiles from ground-based remote sensing. During HOPE, different remote sensing instruments to measure water vapour, both active and passive, were deployed.

An active method is given by the Raman lidar technique (Ansmann et al., 1992; Whiteman et al., 1992; Wandinger, 2005). Water vapour mixing ratio has been determined for several decades using this technique (Melfi et al., 1969; Cooney, 1970;
Melfi, 1972). With advancing technology Raman lidars enabled high vertical resolution measurements of water vapour and extended their range to the whole troposphere (Ferrare et al., 1995; Sherlock et al., 1999; Di Girolamo et al., 2009; Leblanc et al., 2012), during daytime (Renaut et al., 1980; Ferrare et al., 2006) or automatically (Goldsmith et al., 1998; Turner et al., 2002). However, water vapour Raman lidars need to be calibrated with an instrument measuring simultaneously for example a microwave radiometer (MWR) or radiosonde (RS) (Mattis et al., 2002; Madonna et al., 2011; Foth et al., 2015). Until now,
lidars were mainly used as research instruments that did not work unattended or automatically on a routine basis. Another major drawback of Raman lidars is that they do not provide any water vapour information from inside the cloud or above due to the strong signal attenuation, especially in liquid clouds. Hence, these measurements are limited from the surface to the cloud base. Furthermore, daytime measurements are limited in height due to the presence of scattered solar radiation (Turner and Goldsmith, 1999).

Another approach is to use passive remote sensing to sound the thermodynamic state of the atmosphere. Passive microwave radiometry can provide atmospheric water vapour observations with high temporal resolution, but limited vertical information (Solheim et al., 1998; Westwater et al., 2005). However, the integrated water vapour (IWV) can be retrieved very accurately. Microwave radiometers can be operated during all weather conditions except for precipitation (Güldner and Spänkuch, 1999). Like for many remote sensing techniques accurate calibrations are crucial for obtaining precise measurements (Maschwitz
et al., 2013; Küchler et al., 2016).

In contrast to the already presented remote sensing observations water vapour profiles can be measured in-situ using RS (Miloshevich et al., 2006). RS launches are mostly performed by national weather services usually twice a day at special locations. Therefore, both horizontal and temporal resolution of routine measurements are rather low. However, these profiles can serve as reference for remote sensing observations.



As described above, it is a challenge to provide continuous high-resolution water vapour profiles with a single instrument. In recent years, the Leipzig Aerosol and Cloud Remote Observations System (LACROS) (Bühl et al., 2013), installed a combination of ground-based remote sensing systems. The synergy of complementary information from both active and passive instruments can provide a more comprehensive understanding of atmospheric processes (Stankov, 1998; Furumoto et al., 2003;

Bianco et al., 2005; Delanoë and Hogan, 2008). From a combination of radar reflectivities and liquid water path from MWR, Frisch et al. (1998) successfully derived liquid water content ($LWC$) profiles. Han et al. (1997) presented a method based on a Kalman Filter (Kalman, 1960; Kalman and Bucy, 1961) that incorporates current and past measurements followed by a statistical inversion that combines the lidar with the radiometer measurement. The Cloudnet project comprises of a number of algorithms for the continuous analysis of cloud properties by means of remote sensing with lidar, MWR and cloud radar

(Illingworth A. J. et al., 2007). The instruments synergy allows for a continuous evaluation of the representation of clouds in climate and weather forecast models (Sengupta et al., 2004; Hogan et al., 2009; Bouniol et al., 2010). Additionally, the data set enables the development and validation of new cloud remote sensing synergy algorithms.

Löhnert et al. (2004, 2008) developed the so-called integrated profiling technique (IPT) that integrates a ground-based MWR, a cloud radar and a priori information, e.g. from RS. This approach enables the derivation of temperature, humidity and liquid

water content profiles (Ebell et al., 2010) and their associated error estimates. The IPT is based on a variational scheme, also known as optimal estimation (Rodgers, 2000). Cimini et al. (2010) as well as Hewison and Gaffard (2006) used a similar approach as Löhnert et al. (2004) but with background information from a short-range numerical weather prediction model instead of RS climatology.

The synergy of Raman lidar and MWR is beneficial for continuously observing the vertical water vapour distribution. When

both Raman lidar and MWR are measuring collocated and simultaneously, continuous water vapour profiles can be obtained operationally (Ferrare et al., 2006; Adam and Venable, 2007; Adam et al., 2010). However, the Raman lidar needs to be calibrated on a routine basis. A calibration method that is based on the IWV from MWR is suited for this issue (Foth et al., 2015). In previous approaches the total precipitable water from MWR in combination with RS has been used to calibrate the water vapour profiles (Turner and Goldsmith, 1999; Turner et al., 2002). Calibration methods only based on RS (England et al.,

1992; Mattis et al., 2002; Reichardt et al., 2012) are inappropriate for continuous monitoring of the tropospheric water vapour with Raman lidar because of their low temporal resolution.

The aim of this study is to present a two-step algorithm that combines a Raman lidar and a MWR by using an optimal estimation approach. The retrieval can be seen as an extension of the IPT by Löhnert et al. (2009). Barrera-Verdejo et al. (2016) also generated a variational retrieval based on these two instruments. On a first glance, both approaches seem to be

similar, but they are fundamentally different with regard to the optimal estimation method. Barrera-Verdejo et al. (2016) used both, Raman lidar and MWR, as part of the observation vector. Since the water vapour profiles from Raman lidar are strongly disturbed by clouds, they are truncated at the cloud base. In the present work, the truncated Raman lidar profiles are extended to the full height range by using a Kalman Filter in a first step. Then the Kalman-filtered profiles serve as input to the optimal estimation. This approach is based on studies of Schneebeli (2009). Additionally, the focus of the presented work is to routinely

retrieve a continuous time series of water vapour profiles and their error estimates during all non-precipitating conditions.



## 2 Instrumentation

In the framework of the HD(CP)$^2$ initiative HOPE was conducted around Jülich in western Germany during April and May 2013 (Macke et al., 2016). The goal of HOPE was to probe the atmosphere with a specific focus on boundary layer development and the development of clouds and precipitation. Two observatories were set up in addition to JOYCE (Löhnert et al.,

2015). The LACROS site (Wandinger et al., 2012; Bühl et al., 2013) was temporarily built up in Krauthausen which is about 4 km south of JOYCE. Both JOYCE and LACROS observatories are equipped with a set of active and passive remote sensing instruments such as lidars and MWRs which allow the application of the proposed retrieval. Radiosondes were launched at the KIT (Maurer et al., 2016) station in Hambach which is about 4 km away from JOYCE and LACROS. Furthermore, a 120 m tower provide surface meteorological data as pressure, temperature and humidity.

### 2.1 Raman lidar Polly$^{XT}$

At LACROS, the lidar measurements were conducted with the fully automatic portable multiwavelength Raman and polarization lidar Polly$^{XT}$ (Althausen et al., 2009) by the Leibniz Institute for Tropospheric Research (TROPOS). Polly$^{XT}$ measures backscattered light at wavelengths of 355, 532 and 1064 nm and Raman scattered light at 387, 407 and 607 nm wavelengths. From that, water vapour profiles can be determined (Whiteman, 2003; Wandinger, 2005). In the lowermost heights the overlap

of the laser beam with the receiver field-of-view of the bistatic system is incomplete. However, the overlap of both Raman channels is assumed to be identical and for that reason the overlap effect is negligible regarding water vapour measurements. Nevertheless, the lowermost 400 m of the signal ratio are set constant to account for the overlap problem. During daytime, no water vapour measurements can be performed due to the high daylight background and the weak signal from Raman scattering. The Polly$^{XT}$ raw data (30 m and 30 s) is processed and calibrated to mixing ratio profiles as explained in Foth et al. (2015). The

vertical and temporal resolution of the calibrated profiles amounts to 90 m and 5 min to decrease the measurement noise and to retrieve water vapour from higher altitudes. The calibrated water vapour profiles are then used for the proposed retrieval.

An overview of the area of operation and the automated measurement capabilities of Polly systems all over the world is extensively introduced by Baars et al. (2016).

### 2.2 Microwave radiometer HATPRO

The humidity and temperature profiler (HATPRO), built by Radiometer Physics GmbH, Germany, is a passive instrument that measures atmospheric emission at two frequency bands in the microwave spectrum. Seven channels are along the 22.235 GHz $H_2O$ absorption line. From these observations humidity information can be retrieved. The seven channels of the other band from 51 to 58 GHz along the $O_2$ absorption complex contain the vertical temperature profile information. The fully automatic microwave radiometer HATPRO allows to derive temperature and humidity profiles as well as integrated quantities such as

integrated water vapour (IWV) and liquid water path (LWP) with a high temporal resolution up to 1 s (Rose et al., 2005). Their uncertainties amount to $0.5\,\mathrm{kg\,m^{-2}}$ for IWV (Steinke et al., 2015) and to $22\,\mathrm{g\,m^{-2}}$ for low LWP values and increase up to





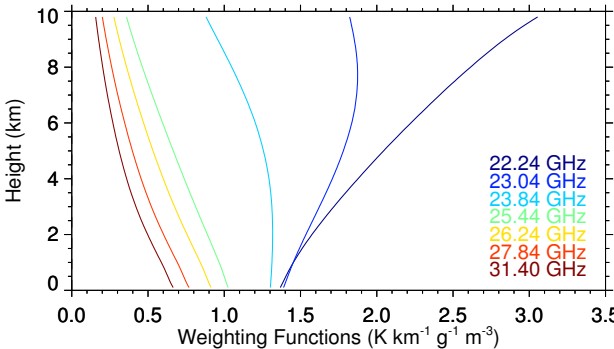

**Figure 1.** Absolute humidity weighting function for the HATPRO frequencies for a cloud free model atmosphere.

$45\,\mathrm{g\,m^{-2}}$ for LWP values higher than $500\,\mathrm{g\,m^{-2}}$, respectively (Ebell et al., 2011). Observations are possible during nearly all weather conditions except precipitation.

Statistical algorithms were used to retrieve temperature profiles, IWV and LWP from the measured brightness temperatures by means of a multi-linear regression between modelled brightness temperatures and atmospheric profiles. That algorithm is based on a long-term dataset of De Bilt radiosondes (Löhnert and Crewell, 2003).

Weighting functions, also called Jacobians, are well suited to describe the ability for humidity profiling. Figure 1 shows the weighting functions for the seven HATPRO frequencies along the $H_2O$ absorption band. Generally, the measured brightness temperatures do not originate from an isolated height level. The weighting functions describe the contribution of a certain height to the observed signal. Ideally, the weighting functions are peaked functions and several frequencies contribute information from different height levels. Three weighting functions (22.24, 23.04 and 23.84 GHz) differ considerably from each other. The higher frequencies have a similar shape as the atmosphere is optically thin at these frequencies. For that reason they add only little information and the vertical distribution of humidity is limited.

The usage of the 31 GHz channel caused unrealistic results. The reason for that behaviour was not identified but might be induced by the forward model or a faulty calibration.

## 2.3 Radiosondes

During HOPE, radiosondes (RS) were launched minimum twice a day (11:00 and 23:00 UTC) and more often during intensive observation periods (IOP) at the KITCube site in Hambach. The RS (type Graw DFM-09) measures temperature, humidity, pressure and wind velocity (Nash et al., 2011; Wang and Zhang, 2008). Due to the vicinity of the RS station to an open-cast mining with a depth of nearly 400 m, horizontal inhomogeneities between the RS launch station and LACROS are likely (Foth et al., 2015).





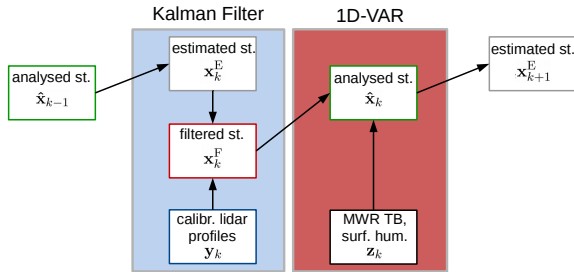

**Figure 2.** Sketch of the retrieval scheme. Details are given in the text. This figure is adapted from Schneebeli (2009).

## 3   Retrieval methodology

The focus of this work is to retrieve a continuous time series of water vapour profiles from a combination of ground-based re-mote sensing with Raman lidar and MWR in a straightforward way to offer a broad application. Most of this section has already been described and presented in Foth (2017) without explicitly citing. The retrieval is a two-step algorithm that combines the

Raman lidar mixing ratio profile with the MWR brightness temperatures. The Kalman filter (first step) eliminates measurement disruptions (e.g. clouds) to provide a full height mixing ratio profile that serves as input to the one-dimensional variational as-similation (optimal estimation method). The retrieval can be applied to raw data (photon counts) using the calibration method based on (Foth et al., 2015) or using already calibrated profiles.

Figure 2 gives a brief overview of the retrieval framework. It starts with the latest analysed state $\hat{x}_{k-1}$ which is projected

in time to the estimated state $x_k^{\mathrm{E}}$ with $k$ being the time index. This state is then combined with the current lidar measurement $y_k$ to the filtered state $x_k^{\mathrm{F}}$ using the Kalman filter. $x_k^{\mathrm{F}}$ is then used as the a priori input to the one-dimensional variational assimilation. The a priori profile is modified such that the modelled brightness temperature match those measured with the microwave radiometer (MWR) $z_k$ resulting in the most probable estimated state $\hat{x}_k$ which is again projected in time in the consecutive step. Inverse methods for atmospheric sounding are well described in Rodgers (2000). For clarity the same notation

is used.

### 3.1   Definition of quantities

In this section the state vector and the two measurement vectors are described. The first measurement vector contains the mixing ratio profile from the lidar measurement. It is used in the first retrieval step (Kalman filter). The second measurement vector consists of the brightness temperatures from the MWR measurement and a surface mixing ratio from a standard meteorological

station. This vector is used in the optimal estimation.

The atmospheric state is described by the state vector

$$\boldsymbol{x} = [q_1, \ldots, q_n]^T \tag{1}$$





which contains the humidity variable $q$ at different height levels from 0 to height $n$ (e.g. $10\,\text{km}$). The vertical resolution originates from the lidar measurements and accounts to $90\,\text{m}$. The humidity variable $q$ is given as the natural logarithm of water vapour mixing ratio. The benefit of using the logarithm is the limited range of variation and the prevention of negative unphysical values (Phalippou, 1996).

The lidar measurement vector of length $m_y$

$$\boldsymbol{y} = [q_1, \ldots, q_{m_y}]^T \tag{2}$$

contains the water vapour mixing ratio at each height level from ground up to a possible cloud base. The lidar profiles $\boldsymbol{y}$ and the associated errors $\boldsymbol{\epsilon}_y$ are usually given in mixing ratio. For the reasons mentioned above, both have to be transformed into $q$ values. The transformed errors define the diagonal elements of the lidar measurement covariance matrix $\mathbf{S}_y$. The off-diagonal

elements are assumed to be zero which means that no correlation exists between the errors at certain height levels.

The second measurement vector, from now on observation vector, is given as:

$$\boldsymbol{z} = [T_{\text{B},1}, \ldots, T_{\text{B},m_\nu}, q_{\text{s}}]^T \tag{3}$$

with the dimension $m_z$. It contains the brightness temperature $\boldsymbol{T}_\text{B}$ at a certain frequency $\nu$ and the surface mixing ratio $q_\text{s}$ from a standard meteorological station. In this study only zenith observations and frequencies along the water vapour absorption

band are chosen. The combined measurement and forward model covariance matrix $\mathbf{S}_z$ contains the errors from the MWR observation, from the surface mixing ratio measurement and from the forward model. The errors from the MWR observation are the radiometric noise. Its variance is set to $0.25\,\text{K}^2$ at each frequency. The off-diagonal elements are set to $0.01\,\text{K}^2$ meaning small covariances between the frequencies. The determination of the forward model error described in Sec. 3.3. Forward model uncertainties that occur due to assumptions in the **LWC** profiles are illustrated in Sec. 3.4. The measurement uncertainty of the

surface mixing ratio amounts to $0.1\,\text{g}\,\text{kg}^{-1}$. However, the uncertainty is increased due to the distance between the observation platform and the surface humidity sensor (see Sec. 2) and is assumed to be $0.3\,\text{g}\,\text{kg}^{-1}$.

First guess profiles and errors are created for the HOPE campaign. Usually they are formed by a certain amount of RS. Therefore the covariance matrix is sometimes called RS climatology. For the HOPE campaign 211 RS were used to calculate a mean profile that serves as first guess profile and is used after a long measurement disruption. Additionally, the correlation

and covariance matrices are determined (Fig. 3). Here, the humidity variable is interpolated to the state grid space (lidar height grid) and is transformed to the natural logarithm before calculating the matrices. Both clearly illustrate the correlations between water vapour at different heights in the atmosphere. Naturally the correlation is close to one near the main diagonal and is smaller for off-diagonal terms. Due to well-mixed conditions the correlation in the lowest $1.5\,\text{km}$ is higher. These matrices are similar to those from previous studies (Ebell et al., 2013; Barrera-Verdejo et al., 2016).

## 3.2   Kalman filter

In the presence of clouds, the lidar profile is truncated at the cloud base due the strong attenuation within the cloud. We use the Kalman filter to expand the truncated lidar profile to the full height range using previous informations. The Kalman filter is




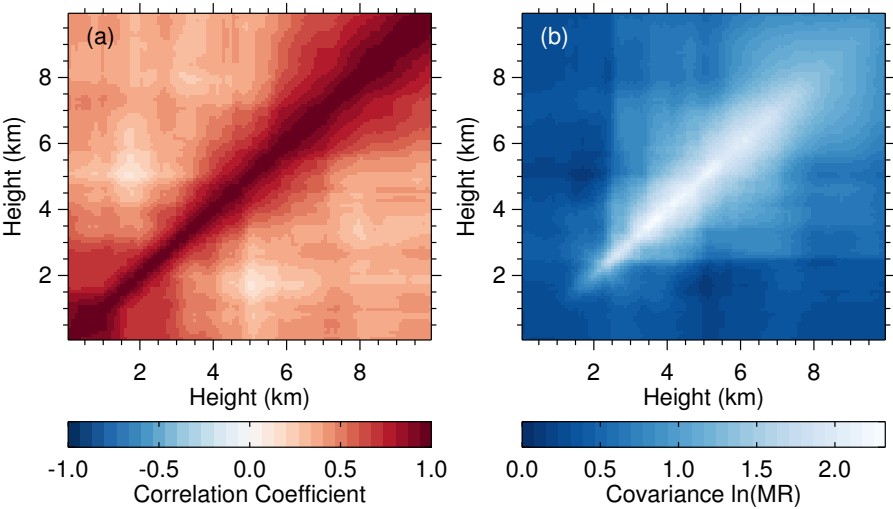

**Figure 3.** Correlation **(a)** and covariance matrix **(b)** derived from 211 radiosondes for HOPE. Both are shown for the natural logarithm of the mixing ratio (ln(MR)) as function of height with a resolution of 90 m.

based on the following two equations:

$$\boldsymbol{y}_k = \mathbf{H}_k \boldsymbol{x}_k + \boldsymbol{\epsilon}_{y,k} \tag{4}$$

$$\boldsymbol{x}_{k+1} = \mathbf{M}_k \boldsymbol{x}_k + \boldsymbol{\epsilon}_{t,k}. \tag{5}$$

The evolution operator (e.g. forward model) $\mathbf{H}_k$ projects the state into measurement space (Eq. 4). Since $\boldsymbol{x}_k$ and $\boldsymbol{y}_k$ use the

same humidity variable, the forward model matrix $\mathbf{H}_k$ equals the unity matrix with dimension $m_y \times n$. Equation (5) describes the transition of the state vector at time step $k$ to time step $k+1$. The transition matrix $\mathbf{M}_k$ is assumed to be the unity matrix due to the lack of an atmospheric model. The transition error $\boldsymbol{\epsilon}_{t,k}$ corresponds to the covariance matrix $\mathbf{S}_{t,k}$. For the calculation of $\mathbf{S}_{t,k}$ the Schneebeli method can be applied (Schneebeli, 2009). He generated a time series of synthetic profiles from a combination of consecutive radiosondes and ground values. $\mathbf{S}_{t,k}$ is finally calculated from an ensemble of these consecutive

profiles. A similar approach is described by Han et al. (1997). After a large number of time steps, it might happen that the correlations between layers get lost. Another possibility is to start with RS climatology covariance as previous covariance matrix ($\hat{\mathbf{S}}_{k-1}$) at every consecutive time step. Using this approach the addition of the transition covariance matrix ($\mathbf{S}_{t,k}$) can be skipped. In this application the latter approach is used.

Using Eq. (5), the last analysed state $\hat{\boldsymbol{x}}_{k-1}$ and its covariance matrix $\hat{\mathbf{S}}_{k-1}$ are propagated as follows:

$$\mathbf{x}_k^{\mathrm{E}} = \mathbf{M}_k \hat{\boldsymbol{x}}_{k-1} \tag{6}$$

$$\mathbf{S}_k^{\mathrm{E}} = \mathbf{M}_k \hat{\mathbf{S}}_{k-1} \mathbf{M}_k^T + \mathbf{S}_{t,k}. \tag{7}$$





$x_k^{\mathrm{E}}$ and $\mathbf{S}_k^{\mathrm{E}}$ are the estimated state and its covariance matrix, respectively. These are then combined with the lidar measurement at time step $k$ to the filtered state:

$$x_k^{\mathrm{F}} = x_k^{\mathrm{E}} + \mathbf{G}_k^{\mathrm{K}} \left[ y_k - \mathbf{H}_k x_k^{\mathrm{E}} \right] \tag{8}$$

with $\mathbf{G}_k^{\mathrm{K}}$ being the Kalman gain matrix:

$$\mathbf{G}_k^{\mathrm{K}} = \mathbf{S}_k^{\mathrm{E}} \mathbf{H}_k^T \left[ \mathbf{H}_k \mathbf{S}_k^{\mathrm{E}} \mathbf{H}_k^T + \mathbf{S}_{y,k} \right]^{-1}. \tag{9}$$

The covariance matrix of the filtered state is determined by:

$$\mathbf{S}_k^{\mathrm{F}} = \mathbf{S}_k^{\mathrm{E}} - \mathbf{G}_k^{\mathrm{K}} \mathbf{H}_k \mathbf{S}_k^{\mathrm{E}}. \tag{10}$$

Finally, $x_k^{\mathrm{F}}$ and $\mathbf{S}_k^{\mathrm{F}}$ servee as input to the optimal estimation.

The application of this technique for linear filtering and prediction problems was first described by Kalman (1960); Kalman and Bucy (1961).

### 3.3 Forward model

In the optimal estimation framework microwave brightness temperatures ($T_{\mathrm{B}}$) at given frequencies ($\nu$) are modelled from the a priori atmospheric profiles and are compared to those that are measured. However, in this work only zenith observations are used. Based on Simmer (1994), $\mathbf{F}(x)$ models the non-scattering microwave radiative transfer using gas absorption by Rosenkranz and liquid water absorption by Liebe (Rosenkranz, 1998; Liebe et al., 1993) for each height level of the retrieval grid (90 m). The Rosenkranz gas absorption model is corrected for the water vapour continuum absorption according to Turner et al. (2009). The humidity information ($q$) of the a priori profile originates from the Kalman filtered state, whereas the temperature profiles ($T$) are provided by statistical retrievals from MWR observations (Sec. 2.2). The pressure profiles ($p$) are calculated by surface pressure observations from MWR and the barometric formula. Since the retrieval grid is limited to 10 km, the thermodynamic state between 10 and 30 km is taken from a RS climatology above Essen which is in the vicinity of the HOPE area. The restriction to the troposphere up to 10 km would lead to errors of around 1 K in the calculation of the brightness temperatures. Assumptions about the liquid water content ($LWC$) and its determination are described in Sec. 3.4. The forward modelling of the surface mixing ratio is trivial. It is a one : one translation to the lowest level of the state vector $\mathbf{x}$. In conclusion $\mathbf{F}(x)$ is of the following form:

$$\mathbf{F}(x) = \begin{pmatrix} \mathrm{RTO}(T,q,p,LWC,\nu_1) \\ \vdots \\ \mathrm{RTO}(T,q,p,LWC,\nu_{m_\nu}) \\ q_1 \end{pmatrix} \tag{11}$$

with RTO being the radiative transfer operator.



**Table 1.** Forward model error for each frequency due to different absorption codes. Uncertainties are given as square root of the diagonal elements of the covariance matrix.

| Channel number | Frequency (GHz) | HATPRO uncertainty (K) |
|---|---|---|
| 1 | 22.24 | 0.07 |
| 2 | 23.04 | 0.2 |
| 3 | 23.84 | 0.42 |
| 4 | 25.44 | 0.56 |
| 5 | 26.24 | 0.55 |
| 6 | 27.84 | 0.53 |
| 7 | 31.40 | 0.51 |

The forward model error is calculated as covariance of the difference between brightness temperatures modelled by two different absorption codes, Rosenkranz and Liebe (Rosenkranz, 1998; Liebe et al., 1991) applied to a longterm data set of radiosondes from Lindenberg, Germany. The diagonal elements of its covariance matrix are shown in Tab. 1. One has to consider that there are significant off-diagonal terms. This error is part of the combined observation and forward model covariance $\mathbf{S}_z$.

The uncertainties of the gas absorption models cause biased mixing ratio profiles (see Sec. 5).

### 3.4 Liquid water assumption

Since liquid water strongly affects the absorption in the microwave spectrum, its amount and height have to be known. However, from MWR only the integral value can be derived, and not its vertical distribution. In order to determine LWC profiles, the cloud boundaries have to be determined. The cloud base of a liquid water cloud is identified by the gradient method based on

the 1064 nm channel from lidar (Baars et al., 2008). In this work, it has been shown that this method is more robust for the automatic detection of the cloud base than the wavelet covariance transform (Brooks, 2003; Baars et al., 2008). However, a threshold value has to be chosen carefully to distinguish between thin liquid water clouds and optically thick aerosol layers below liquid water clouds. Additionally, liquid water clouds are only detected if the LWP is larger than a narrow threshold of $5\,\mathrm{g\,m^{-2}}$.

The $\boldsymbol{LWC}$ is calculated from the modified adiabatic assumption (Karstens et al., 1994):

$$\boldsymbol{LWC} = \boldsymbol{LWC}_{\mathrm{ad}}\left[1.239 - 0.145\ln(h)\right], \tag{12}$$

where $h$ indicates the height above cloud base in m and $h$ within the range of 1 and 5140 m. The adiabatic $\boldsymbol{LWC}_{\mathrm{ad}}$ is calculated using the temperature and pressure profiles and is corrected for effects of dry air entrainment, freezing drops or precipitation. The $\boldsymbol{LWC}$ is integrated over all layers until the calculated LWP equals the LWP measured with MWR. This height is finally

defined as cloud top. However, any profile is treated as single layer cloud with this method.





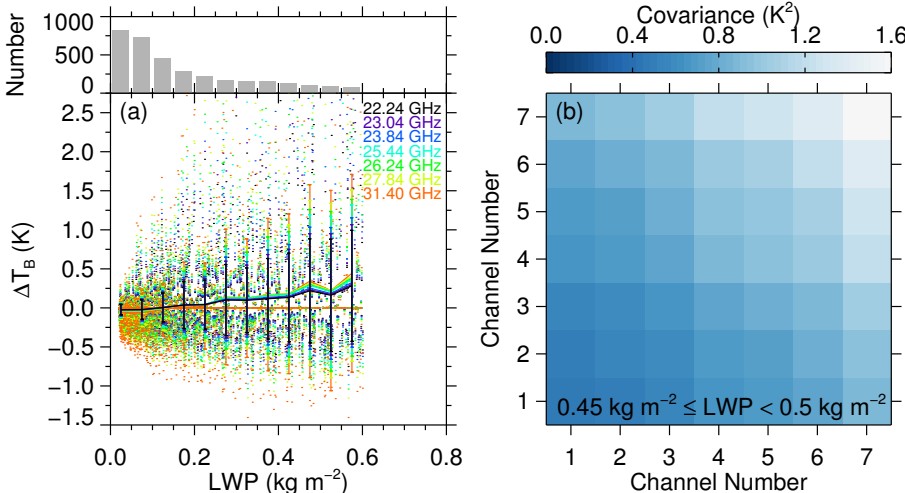

**Figure 4. (a)** Brightness temperature difference as function of LWP (dots) using two different **LWC** assumptions. The colours indicate the according frequencies (top right). The mean and the standard deviation per bin size are indicated by coloured lines and error bars, respectively. The bin size amounts to $0.05\,\mathrm{kg\,m^{-2}}$. The number of occurrences is given in grey bars at the top. **(b)** Exemplary covariance matrix for an LWP between 0.45 and $0.5\,\mathrm{kg\,m^{-2}}$. The channel numbers correspond with the HATPRO frequencies given in **(a)** that means 1 refers to 22.24 GHz etc.

Usual approaches to diagnose $LWC$ profiles from radiosonde are based on a threshold method (Wang et al., 1999). Cloud bases or tops are identified when the relative humidity exceeds or falls below 95 %, respectively. Within the cloud the $LWC$ is calculated using the modified adiabatic assumption (Löhnert and Crewell, 2003). The uncertainty that results in the assumption of single layer clouds is estimated by comparing both mentioned methods. This is done for a long term data set of radiosondes

from Lindenberg, Germany. For these radiosonde profiles brightness temperatures are modelled at the HATPRO frequencies using both $LWC$ profile assumptions. The brightness temperature difference as function of LWP is illustrated in Fig. 4 (a). As can be seen, the means and standard deviations (coloured lines and error bars) increases with increasing LWP. In addition, the difference increases from the 22.24 to 31.4 GHz. Naturally, there is no difference for single layer clouds indicated by the dots at 0 K. The number of occurrences decreases with increasing LWP (grey bars on the top). However, only clouds with

an LWP larger than $0.02\,\mathrm{kg\,m^{-2}}$ are considered. Figure 4 (b) shows an exemplary covariance matrix for an LWP between 0.45 and $0.5\,\mathrm{kg\,m^{-2}}$. These uncertainties contain significant off-diagonal terms and are larger for the channels that are more sensitive to liquid water (31.4 GHz). According to the observed LWP the corresponding covariance is added to the combined observation and forward model covariance matrix $\mathbf{S}_z$ to account for the assumption of single layer liquid water clouds.

## 3.5 Optimal estimation method (OEM)

A schematic overview over the optimal estimation is given in Fig. 5. In basic terms, the forward model simulates what the MWR would observe given an arbitrary state. The problem is that several different states may produce the same measurement.




This is a so-called ill-posed problem. To constrain the state space a priori informations as lidar profiles are needed. In the proposed retrieval the lidar profiles are Kalman filtered as mentioned above. Finally, the optimal estimation finds the most probable solution (mixing ratio profile) from a class of solutions. The theory of inverse modelling based on optimal estimation methods is briefly introduced in this section.

The optimal estimation of an atmospheric state by a given observation vector $z$ and an a priori state $x_a = x^F$ can be found by minimising the cost $J(\hat{x})$ function of the form (Rodgers, 2000)

$$J(\hat{x}) = J_a(\hat{x}) + J_z(\hat{x}) + J_{sup}(\hat{x}) \tag{13}$$

$J_a(\hat{x})$ indicate the a priori costs, $J_z(\hat{x})$ the observation costs and $J_{sup}(\hat{x})$ is a penalty term to avoid supersaturation. Since both liquid and ice phase can occur in clouds at temperatures between $-38$ to -5 °C (Heymsfield and Sabin, 1989; Koop et al.,
2000; Ansmann et al., 2009; Kanitz et al., 2011), the saturation mixing ratio is defined as follows:

$$q^{sat} = \begin{cases} q_{liq}^{sat} & : -5\,°C < \vartheta \\ q_{lin}^{sat} & : -38\,°C < \vartheta < -5\,°C \\ q_{ice}^{sat} & : \vartheta < -38\,°C \end{cases} \tag{14}$$

where $q_{liq}^{sat}$ and $q_{ice}^{sat}$ are the saturation mixing ratios above liquid water and ice, respectively. $q_{lin}^{sat}$ denotes a linear function that describes the transition from $q_{liq}^{sat}$ to $q_{ice}^{sat}$. The according uncertainty is defined as the difference between $q_{liq}^{sat}$ and $q_{lin}^{sat}$ and between $q_{lin}^{sat}$ and $q_{ice}^{sat}$, respectively. It amounts to a maximum of $0.23\,g\,kg^{-1}$ at -8 °C and decreases with decreasing
temperature that usually means increasing height.

    $J_{sup}(\hat{x})$ adds a penalty if the retrieval produces supersaturation all over the profile (Phalippou, 1996; Schneebeli, 2009). This function is defined by

$$J_{sup}(\hat{x}) = \sum_{j}^{n} \boldsymbol{J}_{sup}(x_j) \tag{15}$$

$$\boldsymbol{J}_{sup}(x_j) = \begin{cases} 0 & : q_j \leqslant q_j^{sat} \\ \zeta \left( q_j - q_j^{sat} \right)^3 & : q_j > q_j^{sat}. \end{cases} \tag{16}$$

The constant $\zeta = 10^6$ drives the strictness of the constraint. The larger $\zeta$, the more strict is the constraint. Here, a large value is set, to avoid supersaturation all over the profile. However supersaturation is not completely avoided due to the uncertainties in the temperature profiles from the MWR that are the basis of the saturation mixing ratio $q^{sat}$.

    The implementation of a constraint that prohibits subsaturation within clouds is not beneficial in this application. The assumption of single layer liquid water clouds and the uncertainties in the temperature profile would result in uncertain saturation
mixing ratio profiles and finally lead to wrong retrievals.





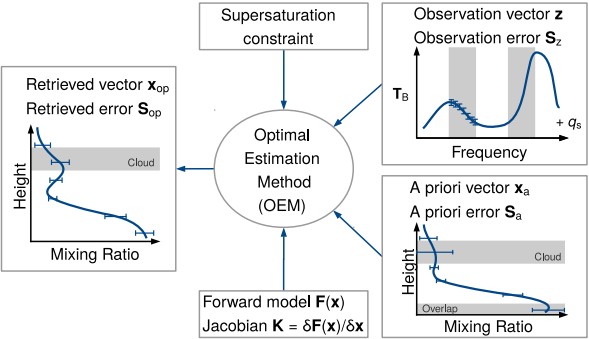

**Figure 5.** Illustration of the optimal estimation method. Details are given in the text.

Written out Eq. (13) becomes to:

$$J(\hat{\boldsymbol{x}}) = [\hat{\boldsymbol{x}} - \boldsymbol{x}_a]^T \mathbf{S}_a^{-1} [\hat{\boldsymbol{x}} - \boldsymbol{x}_a]$$
$$+ [\boldsymbol{z} - \mathbf{F}(\hat{\boldsymbol{x}})]^T \mathbf{S}_z^{-1} [\boldsymbol{z} - \mathbf{F}(\hat{\boldsymbol{x}})]$$
$$+ J_{\text{sup}}(\hat{\boldsymbol{x}}). \tag{17}$$

For clarity the time index is omitted here. $\hat{\boldsymbol{x}}$ is the optimal estimate of the atmospheric state. $\mathbf{S}_a$ and $\mathbf{S}_z$ denote the covariance

matrices of the a priori and the observation, respectively. The optimum solution can be found iteratively using the Levenberg-Marquardt method:

$$\boldsymbol{x}_{i+1} = \boldsymbol{x}_i + \left[ (1+\gamma)\mathbf{S}_a^{-1} + \mathbf{K}_i^T \mathbf{S}_z^{-1} \mathbf{K}_i + \ddot{\mathbf{J}}_{\text{sup}} \right]^{-1}$$
$$\times \left[ \mathbf{K}_i^T \mathbf{S}_z^{-1} (\boldsymbol{z} - \mathbf{F}(\boldsymbol{x}_i)) + \mathbf{S}_a^{-1} (\boldsymbol{x}_i - \boldsymbol{x}_a) + \dot{\mathbf{J}}_{\text{sup}} \right] \tag{18}$$

with $i$ being the iteration index. The dots above $\boldsymbol{J}$ indicate the first and the second derivative, respectively. The Levenberg-

Marquardt parameter $\gamma$ is increased by a factor of 10 if $J(\hat{\boldsymbol{x}}_{i+1}) \geqslant J(\hat{\boldsymbol{x}}_i)$ and reduced by a factor of 2 if $J(\hat{\boldsymbol{x}}_{i+1}) < J(\hat{\boldsymbol{x}}_i)$. In this work the initial value of $\gamma = 2$. It was found that the Levenberg-Marquardt method does not reach convergence faster but more reliably than the Gauss-Newton approach ($\gamma = 0$) (Rodgers, 2000; Schneebeli, 2009). If $\gamma \to \infty$, the step tends towards the steepest descent of the cost function, allowing to leave a local minimum towards a global minimum (Hewison and Gaffard, 2006). $\mathbf{K}_i$ denotes the weighting function matrix, also known as Jacobian or Kernel (hence $\mathbf{K}$), but from now on Jacobian. It

is defined as:

$$\mathbf{K} = \frac{\partial \mathbf{F}(\hat{\boldsymbol{x}})}{\partial \hat{\boldsymbol{x}}} \tag{19}$$

and calculated by perturbing the state vector at each height level by $\ln(0.1\,\mathrm{g\,kg^{-1}})$. Equation 18 is iterated until the following criterion is fulfilled:

$$[\mathbf{F}(\boldsymbol{x}_{i+1}) - \mathbf{F}(\boldsymbol{x}_i)] \mathbf{S}_{\delta z}^{-1} [\mathbf{F}(\boldsymbol{x}_{i+1}) - \mathbf{F}(\boldsymbol{x}_i)] \ll m, \tag{20}$$





with $\mathbf{S}_{\delta z}$ being the covariance matrix between the measurement and $\mathbf{F}(\hat{\boldsymbol{x}})$ :

$$\mathbf{S}_{\delta z} = \mathbf{S}_z \left( \mathbf{K}\mathbf{S}_a\mathbf{K}^T + \mathbf{S}_z \right)^{-1} \mathbf{S}_z. \tag{21}$$

Finally, the covariance matrix of the resulting analysed state vector (a posteriori) is calculated as

$$\hat{\mathbf{S}} = \left( \mathbf{K}^T \mathbf{S}_z^{-1} \mathbf{K} + \mathbf{S}_a^{-1} \right)^{-1}. \tag{22}$$

Since the retrieval might converge at a false minimum it is necessary to check the retrieval for correct convergence. Therefore, the $\chi^2$ test for consistency of the optimal retrieval ($\boldsymbol{x}_{\mathrm{op}}$) with the observation ($\boldsymbol{z}_{\mathrm{obs}}$) is introduced:

$$\chi^2 = \left[ \mathbf{F}(\boldsymbol{x}_{\mathrm{op}}) - \boldsymbol{z}_{\mathrm{obs}} \right]^T \mathbf{S}_{\delta z}^{-1} \left[ \mathbf{F}(\boldsymbol{x}_{\mathrm{op}}) - \boldsymbol{z}_{\mathrm{obs}} \right]. \tag{23}$$

Herein, the forward modelled state $\mathbf{F}(\boldsymbol{x}_{\mathrm{op}})$ and the observation vector $\boldsymbol{z}_{\mathrm{obs}}$ are compared with the error covariance matrix $\mathbf{S}_{\delta z}$. The test is usually used to look for outliers, i.e. cases where the $\chi^2$ value is larger than a threshold value ($\chi_{\mathrm{thr}}$). $\chi_{\mathrm{thr}}$ is

calculated for a probability of 5 % that $\chi^2$ is greater than the threshold for a theoretical $\chi^2$ distribution with $m_z$ degrees of freedom. All retrieved profiles with a $\chi^2$ value that exceeds the threshold are marked as untrustworthy. The $\chi^2$ values of all retrieved profiles are analysed and discussed in Sec. 5.

The averaging kernel matrix $\mathbf{A}$ gives the sensitivity of the retrieval to the true state:

$$\mathbf{A} = \frac{\partial \hat{\boldsymbol{x}}}{\partial \boldsymbol{x}} = \left( \mathbf{K}^T \mathbf{S}_z^{-1} \mathbf{K} + \mathbf{S}_a^{-1} \right)^{-1} \mathbf{K}^T \mathbf{S}_z^{-1} \mathbf{K}. \tag{24}$$

The rows $\boldsymbol{a}_i^T$ of $\mathbf{A}$ are the averaging kernels. In an ideal inverse method, $\mathbf{A}$ would be a unity matrix. Generally the averaging kernels are peaked functions which indicate the smearing of information across multiple levels. In this work, the averaging kernels are no peaked functions, because the MWR observation does not provide enough vertical information. This issue is covered in detail in Sec. 4.1. The averaging kernel has an area $\boldsymbol{a}_{\mathrm{area}}$ which is a measure of fraction that comes from the observation, rather than the a priori. The area of $\boldsymbol{a}_i$ is the sum of its elements and can be calculated as $\mathbf{A}\boldsymbol{u}$ where $\boldsymbol{u}$ is a vector

with unit elements.

The information content of a measurement can be expressed by the degree of freedom ($d$) which is the trace of $\mathbf{A}$. $d$ is a measure of how many independent quantities are measured. One has to consider that the larger the a priori uncertainty, the larger $d$ and the larger the retrieved a posteriori uncertainty (Ebell et al., 2010).

In summary, the retrieval is strongly driven by the a priori uncertainty which constrains the subspace in which the retrieval

must lie. The larger the off-diagonal elements of this covariance, that means the higher the correlations, the smaller is the subspace. For that reason the a priori covariance has to be estimated very carefully. In the proposed retrieval the a priori covariance is strongly decreased by the application of the Kalman filter that reduces the subspace of possible solutions.





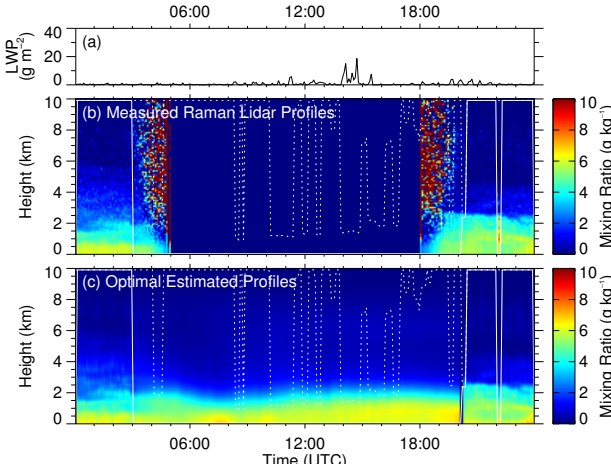

**Figure 6.** Overview of a mostly cloud free case on 5 May 2013. **(a)** liquid water path (LWP). **(b)** Height-time display of the mixing ratio measured by the Raman lidar. **(c)** Height-time display of the retrieved optimal estimated mixing ratio. The solid line indicates the height where the Raman lidar profiles are truncated. The dotted line defines the cloud base height determined by the lidar.

## 4 Retrieval application

### 4.1 Cloud free conditions

In this section the general functionality of the retrieval of water vapour profiles and basic parameters such as averaging kernels and degree of freedom are introduced using a straightforward cloud free case. Figure 6 gives an overview of a mostly cloud free
5    day (5 May 2013). It shows the LWP, the height-time display of the mixing ratio measured by the Raman lidar Polly$^{XT}$ and the height-time display of the retrieved profiles after applying the two-step algorithm. The vertical and temporal resolution of the Raman lidar mixing ratio profiles amounts to 90 m and 5 min, respectively. In the early morning up to 03:00 UTC the mixing ratio could be measured very well by the lidar (Fig. 6 b). With the rising sun the profiles are more and more noisy such that even the lowermost values are disturbed. For that reason the lidar profiles can not be used at all anymore, they serve as an input to the
10    OEM only if they are available. At 05:00 UTC the water vapour channel is automatically switched off and usually switched on again at 18:00 UTC. The noise decreases after sunset allowing an undisturbed water vapour observation from 20:00 UTC on. An automated depolarization calibration produces a gap around 22:00 UTC. The cloud base height indicates the development of boundary layer clouds which can also be seen in the LWP values during daytime (Fig. 6 a). Although there are no lidar profiles during the day, a complete time series of mixing ratio profiles can be retrieved (Fig. 6 c). In the following, the retrieval
15    application of two different conditions, with full height and without mixing ratio profiles from lidar, are distinguished.

Figure 7 illustrates the algorithm processing in the presence of full height calibrated Raman lidar profiles on 5 May 2013, 23:02 UTC. The last analysed state (from 5 min ago) is propagated in time to the estimated state (Fig. 7 a). The propagation is an 1 : 1 translation. Its uncertainty is small because it originates in the last analysed state that was also driven by a lidar





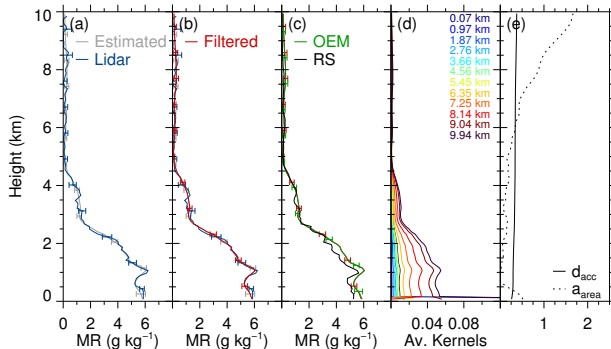

**Figure 7.** Overview of cloud free scene on 5 May 2013, 23:02 UTC. Mixing ratio (MR) profiles from the Raman lidar and the estimated **(a)**, the Kalman filtered **(b)** and the optimally estimated state **(c)**. Additionally, the mixing ratio of the radiosonde (RS) is shown **(c)**. Error bars are added to the profiles at the different states of the processing. **(d)** Averaging kernel for a subset of ten levels indicated by the coloured numbers. **(e)** Accumulated degree of freedom $d_{acc}$ (solid) and the area of the averaging kernel $\mathbf{A}_{area}$ (dotted).

profile. The plotted uncertainties are the square roots of the diagonal elements of the according covariance matrix. The Kalman filter combines the current lidar measurement and the estimated state to the filtered state that is more driven by the estimated state than by the lidar measurement (Fig. 7 b). The filtered profile serves as input (a priori) to the optimal estimation (Fig. 7 c). The small uncertainties of the a priori forces the retrieval to resemble the filtered state with similar uncertainties. The precise

vertical information from the lidar results in small differences to the RS that is used as reference. The comparison to RS is discussed in detail in the next paragraph. Figure 7 (d) shows the averaging kernels for a subset of ten levels. They demonstrate how the information in one retrieved bin is derived from an average of those around it. Ideally the averaging kernels are peaked functions. However, the vertical humidity information at the HATPRO frequencies is limited, which results in smooth functions that are similar to each other. The area of the averaging kernels $a_{area}$ describes the sensitivity to a unit perturbation.

It gives an indication where the MWR observation is sensitive to the true state and where the final information originates. $a_{area}$ values around unity or differing from unity indicate that the information originates in the observation ($z$) or in the a priori, respectively. In Figure 7 (e), $a_{area}$ is close to zero up to 6 km and increases to values around 1.8 in higher altitudes. This means that the MWR observation is not sensitive to the true state, caused by small a priori (Kalman filtered) uncertainty. In this case the retrieved profile is driven by the accurate a priori that originates in the lidar measurement. The information content that

comes from the observation is given by the degree of freedom $d$. Figure 7 (e) represents the accumulated degree of freedom $d_{acc}$ which maximally amounts to $\sim 0.4$. That means that $0.4$ independent pieces of information are added by the observation (MWR and surface value).

As mentioned above, the retrieved optimal profile (OEM) fits well with the RS profile. A more intense comparison is illustrated in Fig. 8 (a). Instead of feeding the retrieval with lidar data, one can only use the MWR as well. In this way, the

improvement of applying Kalman filtered lidar profiles as a priori is emphasized. In such cases (OEM$_{MWR}$) the Kalman filter is completely skipped. The according profile with $d = 2$ is added to Fig. 8 (a). The uncertainties are larger over the whole profile





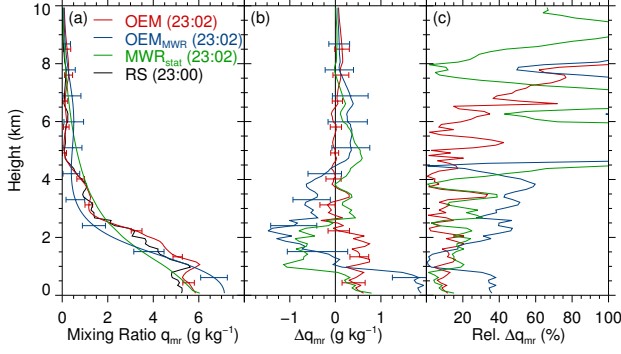

**Figure 8. (a)** Comparison of mixing ratio profiles on 5 May 2013 around 23:00 UTC: retrieved profile (OEM, red), retrieved profile with RS climatology as a priori (OEM$_{MWR}$, blue), profile from the MWR statistical retrieval (green) and RS (black) as reference. Error bars are added to the optimally estimated profiles (red, blue). Absolute **(b)** and relative **(c)** difference from the reference RS.

in comparison to the OEM. Both, the OEM$_{MWR}$ and the MWR profiles from the statistical retrieval (MWR$_{stat}$) are not able to distinguish vertical structures as indicated by the OEM and RS. For that reason, their absolute differences to the RS are larger than those from the OEM (Fig. 8 b). Furthermore, in this application the OEM$_{MWR}$ clearly overestimates the humidity below 1 km. The OEM profile fits best and the zero line (no difference) is within the error bars nearly over the whole profile. The
relative differences (to RS) are smaller below 4 km and is large in altitudes where the mixing ratio from RS is small (Fig. 8 c). In summary, the OEM profile fits best with small uncertainties and differences referred to the RS. However, in cases with full height lidar profiles the optimal estimation is not necessary, because the Raman lidar profiles are already containing nearly all information. But full height lidar profiles are only available 18 % of the time during HOPE and by applying the OEM the dataset is extended to 60 % coverage (see Sec. 5).
In contrast to 23:02 UTC there is no mixing ratio profile from lidar available at 07:02 UTC (Fig. 9 a). Due to the missing lidar profiles the estimated and the filtered profiles as well as their uncertainties are the same (Fig. 9 b). The difference between the filtered and the optimal estimated profile is very small since the atmospheric changes within a 5 min step are quite small. However, the uncertainty decreases near the ground. This is not only caused by the MWR but by the surface measurement which is also part the observation vector ($z$). The optimally estimated profile is very smooth, since the HATPRO frequencies
do not provide enough information to distinguish fine vertical structures. This can be seen in the difference between the optimal estimated profile and the RS profile which is used as reference. The according averaging kernels (Fig. 9 d) are smooth functions that are similar to each other, because the vertical humidity information at the HATPRO frequencies is limited. The area of the averaging kernels $a_{area}$ is around unity (Fig. 9 e). This means that the MWR observation is sensitive to the true state and most information (nearly all) originates in the observation ($z$). The accumulated degree of freedom $d_{acc}$ maximally amounts to $\sim 1.9$
meaning that 1.9 independent pieces of information can be retrieved. Löhnert et al. (2009) used RS climatology as a priori for different locations and found $d$ values around 2 for humidity profiling with HATPRO. In contrast, one has to consider that





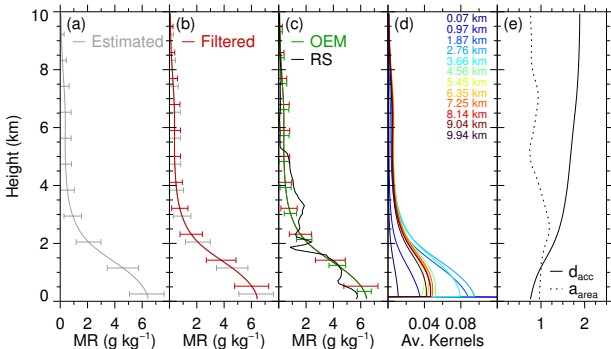

**Figure 9.** As Figure 7, but on 5 May 2013 07:02 UTC.

here the observation vector is supplemented by the surface humidity which also adds information. The difference might be explained by different a priori covariance matrices $\mathbf{S}_a$.

In summary, the presence of a lidar measurement results in more accurate retrievals compared to RS, whereas retrievals without water vapour profiles from lidar are mainly driven by the MWR observation for example during daytime. However, the two-step algorithm allows to retain structures from high vertically resolved lidar data to periods without lidar data.

### 4.2 Cloudy conditions

As introduced in section 3.4, liquid water strongly affects the absorption in the microwave region. Therefore, the operation of the retrieval in the presence of clouds containing liquid water has to be treated separately. Figure 10 shows an overview of a cloudy day, 21 April 2013. In the course of the day the LWP increases to a maximum of $600\,\mathrm{g\,m^{-2}}$ (Fig. 10 a). Between 00:00 and 03:30 UTC the measured lidar profiles reach from ground up to the cloud base between 2.5 and 3.5 km. Referring to the rather low LWP the cloud seems to be an ice cloud. During the day, the mixing ratio is determined on the basis of the MWR observation only disturbed by five short interruptions that are caused by missing cloud base detection by lidar. From 19:30 UTC on the lidar profiles are truncated at the cloud base at around 1.5 km. The LWP shows that these clouds contain liquid water. The possible content of ice water is not relevant for the radiative transfer in the considered spectrum. However, ice clouds as well as all other clouds disturb the precise determination of water vapour with Raman lidar. For that reason the profile is only considered up to cloud base. The problem of truncated profiles is solved by the application of the Kalman Filter (Sec. 3.2). It enhances the profiles up to 10 km by the combination of previous information and the according truncated lidar profile such that a full height profile can serve as input to the optimal estimation.

A comparison between the retrieved profiles (OEM), the retrieved profiles based on climatology ($\mathrm{OEM_{MWR}}$), the MWR profiles from the statistical retrieval ($\mathrm{MWR_{stat}}$) and the RS is shown in Fig. 11 (a). There is a cloud with $\mathrm{LWP} = 242\,\mathrm{g\,m^{-2}}$ between 1.3 and 2.4 km. Both, $\mathrm{OEM_{MWR}}$ and $\mathrm{MWR_{stat}}$, are not able to distinguish the vertical structure inside the cloud given by the RS. Furthermore, they show large differences to the RS profile below and slight above the cloud (Fig. 11 b).





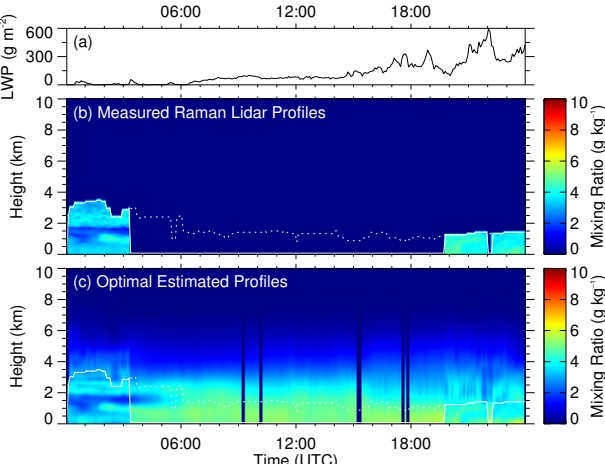

**Figure 10.** As Figure 6 but on 21 April 2013

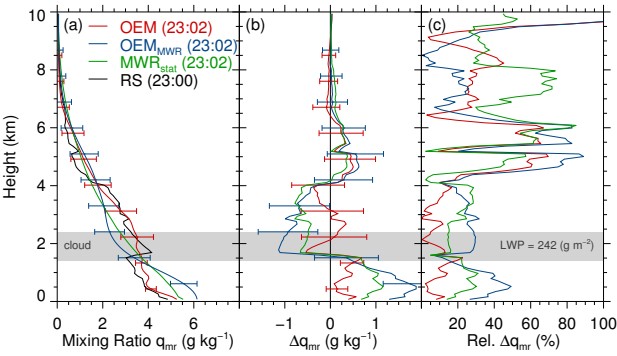

**Figure 11.** As Figure 8 but on 21 April 2013. The grey area indicates the cloud with an LWP of $242\,\mathrm{g\,m^{-2}}$.

The OEM profile shows a good agreement with the RS profile below the cloud based on available lidar data. The associated uncertainties are small. Within the cloud the uncertainty increases. The profile approximates to the RS. Above the cloud, the OEM uncertainties are in the same range than the $\mathrm{OEM_{MWR}}$ profile, whereas the difference to the RS profile is smaller. Nearly over the whole range the RS profile is within the uncertainty range of the OEM profile. The according relative differences to

5    the RS profile are plotted in Fig. 11 (c). Up to 4 km the relative difference of the OEM profile is less than 25 %. Above this height the relative difference increases. The $\mathrm{OEM_{MWR}}$ and $\mathrm{MWR_{stat}}$ have larger relative differences to the RS. In summary, the OEM fits best the RS with lowest differences in and above the cloud.





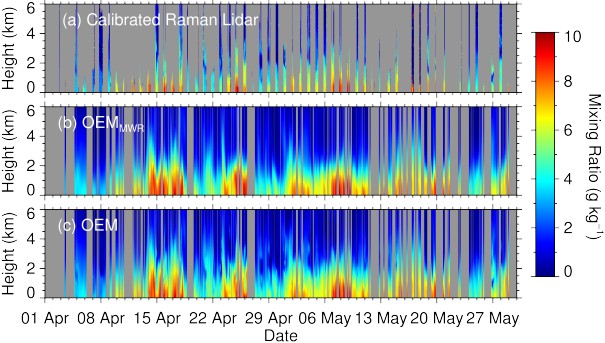

**Figure 12.** Three different Height-time displays of mixing ratio profiles during HOPE: **(a)** calibrated Raman lidar profiles, **(b)** optimal estimated profiles only based on MWR (and surface humidity) without any Raman lidar mixing ratio profile (OEM$_{MWR}$) and **(c)** optimal estimated profiles based on Kalman filtered Raman lidar mixing ratio a priori profiles (OEM).

## 5   Statistical analysis

In the previous section (Sec. 4) the functionality of the retrieval is introduced based on clear sky and cloudy cases during HOPE. A statistical analysis of the retrieved water vapour profiles during the whole HOPE campaign is presented in the following section. Herein also profiles from RS and the OEM$_{MWR}$ (without lidar) are used as reference.

First, an overview over the calibrated water vapour profiles observed by Polly$^{XT}$ during HOPE is given in Fig. 12 (a). The grey area indicates regions without lidar data (up to 6 km) due to cloud attenuation (17 %) and during the day (65 %). The well resolved vertical profiles enable the determination of distinct water vapour structures or inversions that can be seen e.g. at around 1 km in the night between 26 and 27 May 2013.

As introduced in the previous sections, one can use the covariance of the RS climatology as uncertainty from the previous state, instead of lidar data. However, the cloud base height determined by the lidar is necessary. This approach (OEM$_{MWR}$) is only based on the observation with MWR and surface humidity and is similar to that proposed by Löhnert et al. (2009). The according height time display is illustrated in Fig. 12 (b). The gaps (40 %) are caused by rain, MWR breakdowns, flagged MWR data, the absence of cloud base height from lidar or that no solution was found by the retrieval. Nevertheless, the profile availability amounts to 60 %. Although, the data coverage is larger as for Raman lidar (Fig. 12 a), but the vertical resolution is much coarser. This can be seen clearly by comparing to the lidar profiles (Fig. 12 a) in the night between 26 and 27 May 2013.

Figure 12 (c) shows the retrieved mixing ratio profiles (OEM) based on the method that was described in the previous sections. The data coverage is nearly the same as for OEM$_{MWR}$. However, the OEM is able to retrieve fine water vapour structures by means of the lidar profiles. The OEM not only enables the distinction between dry (e.q. beginning of April) and more humid (e.q. middle of April) periods but also the vertical distribution of water vapour especially from within and above a cloud.

For a more comprehensive investigation of the quality of the profiles a differentiation between three situations based on certain initial conditions is helpful. These situations are in accordance with the case studies presented in the previous section





**Table 2.** Overview over the different situations depending on Raman lidar mixing ratio (RL MR) profile availability and truncation height ($h_{tr}$) where the RL MR profile is truncated (due to clouds). The three columns on the right indicate the sample size used for the comparison with radiosonde (RS), to validate the retrieved profiles, and all cases. Furthermore, the profiles that are used for the comparison with RS are separated between those passing and failing the $\chi^2$ test based on a threshold $\chi^2_{thr}$. The temporal resolution of the retrieved profiles amounts to 5 min.

|  | RL MR profiles | Truncation height | Sample size | | |
| --- | --- | --- | --- | --- | --- |
|  |  |  | Comparison RS | | All |
|  |  |  | $\chi^2 < \chi^2_{thr}$ | All | |
| Full height | yes | $h_{tr} > 8\,\mathrm{km}$ | 102 | 131 | 665 |
| Truncated | yes | $0\,\mathrm{km} < h_{tr} \leq 8\,\mathrm{km}$ | 262 | 291 | 2010 |
| No lidar | no | $h_{tr} = 0\,\mathrm{km}$ | 1033 | 1053 | 5732 |
| $\mathrm{OEM_{MWR}}$ | no | — | 1397 | 1475 | 8407 |

(Sec. 4). The first situation includes cases where a full height lidar profile is available (minimum up to 8 km). Such a case is presented in Sec. 4.1 especially in Fig. 7. Referring to the statistical analysis these profiles are marked in blue unless stated otherwise. The second group includes cases with lidar profiles which are truncated between 0 and 8 km mostly due to clouds. Such a case is already introduced in Sec. 4.2 in Fig. 11 and are marked in green from now on. The last group contains all cases

without lidar profiles as introduced in Fig. 9. An overview is given in Tab. 2. The table also lists the sample size for all profiles and those that are used for comparisons with RS. These are also distinguished between profiles passing and failing the $\chi^2$ test that is discussed later in this section. Additionally, the $\mathrm{OEM_{MWR}}$ is used as reference and is marked in grey.

    For assessing the accuracy of a water vapour profile, reference profiles from RS and $\mathrm{OEM_{MWR}}$ profiles are used. In this work the bias and the root mean square error (RMSE) between the retrieved profiles and those from RS are applied to evaluate

the quality of the retrieved profiles. For this comparison retrieved profiles that are between RS launch time and one hour after launch time are used. This results in maximum 12 profiles for one sounding. Only cases which pass the $\chi^2$ test are considered for the comparison. Figure 13 (a) shows the bias for the specified situations and for the $\mathrm{OEM_{MWR}}$. The blue line illustrates the retrieved profiles that are based on lidar profiles in minimum up to 8 km (clear sky). It has a maximum value of $0.5\,\mathrm{g\,kg^{-1}}$ near the surface and it decreases close to zero above 1.5 km. However, the bias is positive that means that the retrieved profiles

have larger values than the RS profiles. Above 6 km the retrieved profiles show higher values than the RS. This issue might be caused by slightly to small determined Raman lidar calibration factors resulting in too small lidar mixing ratios. In these cases, the modelled brightness temperatures for the lidar profiles differ from those measured by MWR. Basically, the uncertainty of Kalman filtered lidar profiles increases with height. This means that the retrieval prevents an increase of the mixing ratio in lower heights due to its small uncertainty. Hence the retrieval tends to overestimate the mixing ratio in larger heights to

minimize the difference between the modelled and the observed brightness temperatures.





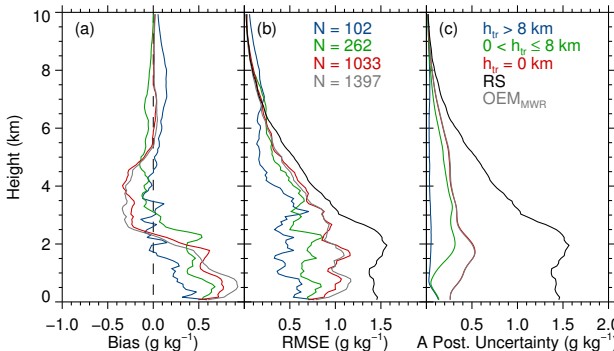

**Figure 13.** Statistical analysis of the synergy improvement: bias to RS **(a)**, root mean square error (RMSE) to RS **(b)** and a posteriori uncertainty **(c)**. It distinguishes four situations according to Tab. 2. The sample size is given by the numbers in the middle panel. Only profiles between RS launch time and one hour after are considered.

The bias of the situations where the lidar profiles are truncated below 8 km is shown in green (Fig. 13 a). The values are in maximum around $0.6\,\mathrm{g\,kg^{-1}}$ and are largest in the PBL. Above 2.5 km the bias is around zero. The bias of the situations where no lidar profiles are available and of the $\mathrm{OEM_{MWR}}$ show a similar behaviour to each other. Both curves show an overestimation of the retrieved mixing ratio within the boundary layer up to 2 km. Between the 2 and 5 km the retrieval underestimates

the mixing ratio by around $-0.4\,\mathrm{g\,kg^{-1}}$. Additionally, the small amount of vertical information that comes from the MWR observation might not be able to compensate this misbehaviour and to resemble the profile given by the reference. This effect can also be seen in the presented clear sky case study in Fig. 9. Nevertheless situations where no lidar profiles are available show a bias closer to zero than the $\mathrm{OEM_{MWR}}$. These cases benefit from the night cases whose vertical structure is propagated into the day cases. The positive biases of all four curves seem to have a systematic difference that might be explained by some sources

of uncertainty in the RS profiles. The different locations of the platform in Krauthausen and the RS launch station and drifts of the balloon might result in the observation of different air masses (Foth et al., 2015). Additionally, RS can have a dry bias (Miloshevich et al., 2001). Naturally, the forward model itself is a source of uncertainty. The modelled brightness temperatures strongly depend on the assumed absorption line shapes (Turner et al., 2009). Figure 14 illustrates a comparison of forward models using two different gas absorption models, Rosenkranz (1998, R98) and Liebe (1993, L93), (Rosenkranz, 1998; Liebe

et al., 1993). Both models are corrected for water vapour continuum absorption according to Turner et al. (2009). All other parameters, e.g. cloud absorption, are the same. Both forward models were performed under two different a priori states, both without lidar. The first uses the a priori profile and the a priori covariance from RS climatology. It simulates the theoretical uncertainty (theor.) only induced by the different absorption models. In the other case the a priori profile is propagated (prop.) from the previous state as used in the original retrieval. Herein, the a priori uncertainty is also taken from the RS climatology.

The bias to RS in the second case is larger since the theoretical uncertainty is propagated from each previous state resulting in an increase of uncertainty (Fig. 14 a). It can be seen that the L93 model has a smaller bias below 1 km. Above 2.5 km the R98 model simulations better fit the RS with a bias around $-0.3\,\mathrm{g\,kg^{-1}}$ and a bias close to $0\,\mathrm{g\,kg^{-1}}$ above 5 km. The retrieved





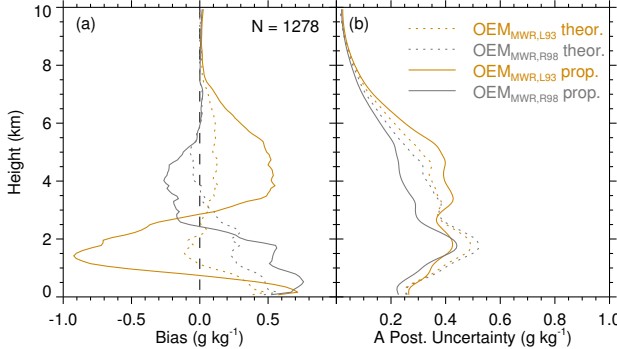

**Figure 14.** Bias to RS **(a)** and a posteriori uncertainty **(b)** for two different absorbtion codes, Rosenkranz, (R98, grey) and Liebe (L93, orange). The shown retrievals are only based on MWR but with different a priori states. On the one hand, both a priori profile and a priori uncertainty are taken from the RS climatology (theor.) and on the other hand the a priori profile is propagated (prop.) from the previous step while the uncertainty is taken from the RS climatology (red cases in the figures above). The sample size is given by the numbers. Only profiles between RS launch time and one hour after launch time are considered.

uncertainty, the so-called a posteriori uncertainty, of the R98 simulations are smaller than those from the L93. The uncertainty of the L93 runs is also largest in heights above 3 km. Finally, the R98 gas absorption model seems to be more suitable for the presented retrieval. Nevertheless, the forward model is a major source of uncertainty.

The RMSE between OEM and RS is illustrated in Fig. 13 (b). It gives an indication about the statistic error. The RMSE of all

four curves decreases with height. In addition, the RMSE is smaller for cases with lidar profiles as a priori and larger for those without. The RMSE of the HOPE RS profiles is larger than any RMSE of the retrieved profiles, that is basically the variance of mixing ratio in the whole period.

Figure 13 (c) illustrates the a posteriori uncertainty of the mixing ratio profiles (see Eq. 22). The black line indicates the uncertainty of the RS climatology that is the square root of the diagonal elements of its covariance matrix. It can clearly be

seen that the retrieved a posteriori uncertainty is smaller for all situations. The curves of the cases without lidar profiles and the $OEM_{MWR}$ are nearly in agreement. In both cases the Kalman filter is skipped due to the absence of lidar profiles. Therefore both use the same a priori uncertainty and their retrievals are solely driven by the MWR and surface humidity observation. The presence of lidar data (full height or truncated) results in much lower uncertainties. Their small a posteriori uncertainties underline the synergy improvement.

In summary, Fig. 13 clearly shows that the application of Kalman filtered lidar profiles enormously improves the accuracy and quality of the retrieved mixing ratio profiles.

Another possibility to evaluate the accuracy of the retrieved profiles is to analyse the bias as function of the mixing ratio (Fig. 15). The slope of the regression line is smaller than the one : one line. This means that, larger differences occur rather for larger mixing ratios. Figure 15 also indicates the correlation between retrieved and RS mixing ratios. The squared coeffi-

cient of correlation $R^2$ is largest for those situations with full height lidar profiles and amounts to 0.97 (Fig. 15 a). The $R^2$ of the





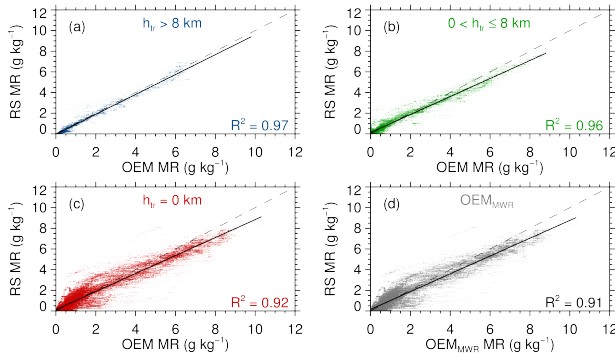

**Figure 15.** Comparison of optimal estimated (OEM) and radiosonde (RS) mixing ratio profiles for four situations given in Tab. 2. The black solid line indicates the regression line.

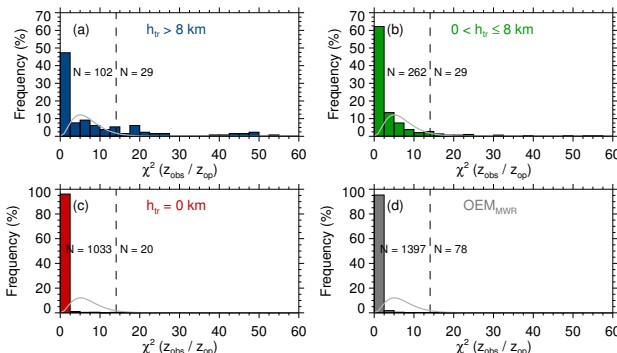

**Figure 16.** Histograms of the $\chi^2$ test for four situation given in Tab. 2. The dotted lines indicate the theoretical $\chi^2$ distribution with $m_y$ degree of freedom. Dashed lines indicate the 5 % threshold value of 14. The absolute number of cases below and above the threshold value is given to the left and to the right side of the dashed line, respectively.

OEM based on truncated lidar profiles (b) is slightly smaller (0.96). In situations without lidar data and the OEM$_{MWR}$ have still smaller values of 0.92 and 0.91, respectively. Nevertheless, all cases show a better agreement with RS than the OEM$_{MWR}$. This illustration also demonstrates the synergy improvement by implementing the lidar data with a Kalman filter before applying the OEM.

5   To assess the quality of retrieved profiles a statistical test for correct convergence of the solution is applied. The modelled state $\mathbf{F}(\boldsymbol{x}_{op})$ and the observation vector $\boldsymbol{z}_{obs}$ are compared with the error covariance matrix $\mathbf{S}_{\delta z}$ (see Eq. 21) to check if the retrieval is consistent with the observation. Figure 16 shows the $\chi^2$ test statistics for all mentioned situations. The $\chi^2$ test was introduced in Sec. 3.5. It can be seen that 29 profiles are rejected in the situations with full height lidar profiles because their $\chi^2$ value exceeds the 5 % threshold value of 14 (Fig. 16 a). The amount of untrustworthy profiles is similar to the situations

10  with truncated lidar profiles. In both cases the smaller a priori uncertainty prevents an adjustment of the modelled brightness temperatures to those measured by MWR. For that reason, their difference is larger resulting in a larger $\chi^2$ value. The $\chi^2$ test





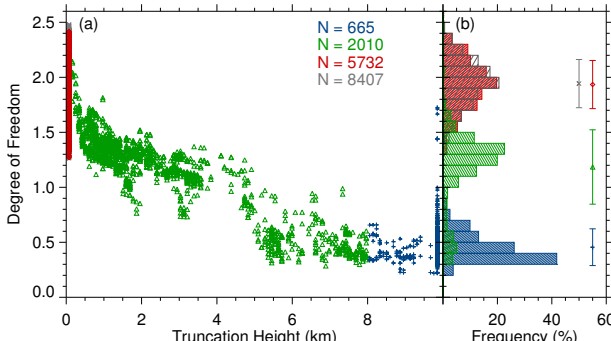

**Figure 17. (a)**: Degree of freedom as function of truncation height for different situations introduced in Tab. 2. **(b)**: Frequency distribution of the degree of freedom. The symbols and error bars correspond to the according mean and standard deviation, respectively. The numbers indicate the sample size of the considered profiles, full height (blue), truncated (green), no lidar (red) and OEM$_{\mathrm{MWR}}$ (grey).

rejects a smaller relative amount of profiles for the daytime cases (c) and at the OEM$_{\mathrm{MWR}}$ (d). Their larger a priori uncertainty enables a better match between the modelled and the measured brightness temperatures. However all situations show a peak at small values that originates in a very good agreement between the forward modelled optimal state and the observation vector. Admittedly the test is very strict and rejects all failing profiles although they might be realistic atmospheric states. Nevertheless,

it enhances the confidence of the retrieved profiles.

A good measure for the proportion of information that comes from the observation is given by the degree of freedom. It describes the number of independent pieces of information that is added by the retrieval and has already been introduced in Sec. 3.5 and 4. Figure 17 (a) illustrates the degree of freedom as function of truncation height. It clearly demonstrates that the lower the truncation height the higher the degree of freedom. This is caused by the larger a priori uncertainty in cases with

truncated or without lidar mixing ratio profiles. The sample size is much higher than in the comparisons above because here all profiles can be used and not only those around the RS launch time. Most of the grey crosses are not visible because they are covered by the red diamonds. The according frequency distributions are shown in Fig. 17 (b). Both the OEM$_{\mathrm{MWR}}$ and the daytime cases are very similar to each other. Even their mean values and standard deviations are nearly identical with values of $1.9 \pm 0.22$. These values are in good agreement with those found by Löhnert et al. (2009) for a similar approach. The situations

with the truncated lidar profiles show a wide range of values from 0.3 to 2.1. The green distribution also has the largest standard deviation which amounts to 0.34. The situations with full height lidar profiles have the smallest mean and standard deviation with values of $0.45 \pm 0.17$. These cases are mostly driven by the a priori information and not by the observation. The variation within each situation is caused by different atmospheric conditions. Figure 18 illustrates the degree of freedom as function of IWV. It shows an increase of $d$ with increasing IWV caused by a stronger emission of water vapour. For higher IWV, the MWR

is able to add more information to the retrieval. Finally, the behaviour of the degree of freedom and especially its dependence on truncation height and hence a priori uncertainty agrees well with similar studies (Löhnert et al., 2009; Ebell et al., 2013).





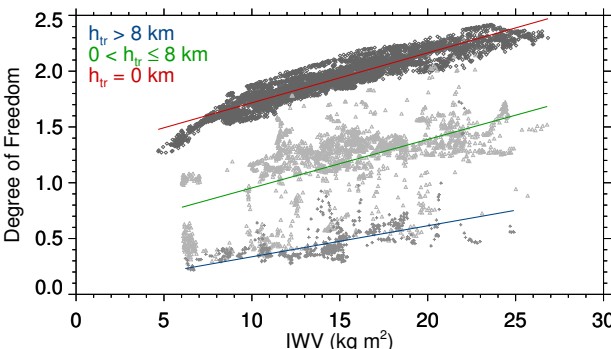

**Figure 18.** Degree of freedom as function of IWV for the situations introduced in Tab. 2. The lines indicate the according regression lines.

## 6 Conclusions

A good knowledge of the water vapour distribution is essential for the description of the thermodynamic state of the troposphere. Since the continuous observation of water vapour profiles with a single instrument is challenging, the synergy of complementary information from active and passive remote sensing became more important in recent years.

In this study we present a two-step retrieval combining the Raman lidar water vapour profiles with the MWR brightness temperatures. The Kalman filtered water vapour profile serve as input (a priori) to the one-dimensional variational approach, also known as optimal estimation. In addition to the water vapour profile, its uncertainty is retrieved.

The retrieval enables the observation of a continuous time series of water vapour profiles with known uncertainties. During HOPE, the availability of full height water vapour profiles from lidar amounts to 17 % excluding all cloudy and daytime cases.

By applying the retrieval, the availability of water vapour profiles can be enlarged to 60 %. The bias with respect to RS and the retrieved a posteriori uncertainty of the retrieved profiles clearly show that the application of the Kalman filter considerably improves the accuracy and quality of the retrieved mixing ratio profiles. In the presence of full height Raman lidar profiles, the MWR does not add much information to the retrieved profiles. However, cases without Raman lidar profiles are dominated by the MWR information with a larger degree of freedom. The lower the truncation height of the lidar profiles the higher the

importance of the MWR.

Furthermore the retrieval can be applied to raw data (photon counts) using the calibration method based on (Foth et al., 2015) or using already calibrated profiles.

In future steps, the precipitation evaporation can be assessed by means of observed or retrieved temperature and humidity profiles. This information can be used to improve model parametrisation of physical processes with water vapour participation

and finally to improve weather and climate predictions.

The retrieval will be implemented into the Cloudnet processing. A better knowledge of the water vapour distribution and the collocated and simultaneous monitoring of cloud microphysics within Cloudnet might improve the understanding of cloud formation, precipitation, evaporation and entrainment rates. The application of this algorithm might help to decrease uncertainties




in the area of cloud and precipitation formation as well as cloud dissipation, as mentioned in the latest IPCC report (Boucher et al., 2013).

*Acknowledgements.* The presented study was conducted within the research programme "High Definition Clouds and Precipitation for Climate Prediction – HD(CP)2 ". This project is funded by the German Federal Ministry of Education and Research within the framework
5    programme "Research for Sustainable Development (FONA)", www.fona.de, under the numbers 01LK1209D and 01LK1504C.

The authors also acknowledge the LACROS team from the Leibniz Institute for Tropospheric Research (TROPOS) for the Raman lidar and microwave radiometer data, the Karlsruhe Institute for Technology (KIT) for the radiosonde launches as well as the Research Center Jülich for the in-situ observations on the 120-m tower.



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
