# Peer review of "Optimal Estimation of Water Vapour Profiles using a Combination of Raman Lidar and Microwave Radiometer"

_Atmospheric Measurement Techniques, 2017_

## Referee Comment (RC1) · Anonymous Referee #1 · 24 May 2017

The paper by Foth et al. presents a retrieval of water vapor from ground based microwave radiometer using an apriori computed from the previous state and collocated lidar measurements. This paper is appropriate for AMT as no science is discussed. Before publication, the paper needs to add a discussion on how the error in the filtered profile changes as the previous state is farther in time. Also, it needs to have a section/examples showing the impact of using the saturation constrain versus not using one.

Comments: P1L17: This sentence implies that the water vapor is some how being dictated by the IPCC, please rewrite. P1L19: Its amount is controlled mostly by the air temperature, rather by emissions. → In the stratosphere, the water vapor is controlled

by the tropopause coldest temperature, do you mean in the boundary layer is controlled by sea surface temperature. Please specify and provide a reference.

P1L24: About the H2O lifetime, is that in the troposphere/boundary layer? Are you sure about this number? Figure 5.23 of "Aeronomy of the Middle Atmosphere: Chemistry and Physics of the Stratosphere (Guy Brasseur and Susan Solomon)" shows a lifetime varying from hundred of years at around 20km to 1 day at around 120km.

P3L6 Please describe what is the difference between this study and Han et al. 1997? Is it just that in here it is used optimal estimation?

P5L6 Weighting functions and jacobians are not the same. Pease double check what are you showing.

P6L9: You are not projecting in time the last analysed state, you are just advancing this state without any modification because H is and M are the unity matrix. Please change

Figure 3: please delete the blue part of the color bar in the correlation coefficient.

Equation 7, why is St,k here, when you just said P8L12 than St,k can be skipped. I would assume that using RS climatology is an extreme case, if you are using the previous state from 5 min ago, the correlation and covariance matrices are going to be completely different to the ones if the previous state were from 6 hours ago (I presume that these ones will look more like the RS climatology) Please clarify.

P10L10. In this work, it has be shown ... This implies that in a previous section of Foth et al 2017 it has been shown that such a method is more robust, which is not the case. Change to: The cloud base of a liquid ... (Baars, et al) which has been shown to be a more robust method for the automatic ...

P12L6: In Rodgers there is no mention of supersaturation cost function. Further, have you try not using such constraint. You are apriori is so tight I do not expect that it is needed.

Equation 18 has an extra bracket. Figure 6 and 10: You could show the OEMMWR to showcase the impact of the lidar apriori.

Figure 8 Could you add the lidar measurement to see if the bias is bigger or smaller than the OEM. Also, is this for 23:02 as in figure 7 or this are for a previous estimate. Further, when you do these comparisons do you apply the averaging kernels, all the MWR will look poorly if you do not include them. Also MWRstat never has an error bar, please add. (also in Figure 11)

Figure 11: What time?

P19L6: For the most part, the RS profile is also within the OEM MWR uncertainty

P21L5: add shown in red after "as introduced in Fig.9"

P21L15-20: Are you implying that the lidar retrieval have a bias for altitude higher than 6km. Could you relax your apriori at those heights so that the information arises from the MWR. This needs to be fixed because in the clear sky the bias should be less than in the rest of the scenarios.

Is the accuracy error shown in Figure 14 taken into account in the error characteristics shown in the previous figures. Also, what are the main differences between L93 and R98, do they use different absorption cross sections, if so, which ones. Speaking of accuracy, what is the impact of the temperature error upon your water vapor retrievals. Would it be better to use a profile derive from RS than from MWR due to the low vertical resolution of the later.

P23L15: enormously is a strong word change to marginally improves.

---

## Referee Comment (RC2) · Anonymous Referee #2 · 1 Jun 2017

Review of "Optimal Estimation of Water Vapour Profiles using a Combination of Raman Lidar and Microwave Radiometer" by Foth and Pospichal

This paper presents an optimal estimation technique for retrievals of water vapor mixing ratio profiles by combining Raman lidar (RL) and microwave radiometer (MWR) data. The paper is well organized, and well written for the most part. I believe it represents a valuable contribution, but I found some of the text and mathematical details a bit difficult to follow. I have provided a number of comments and suggestions that I feel may help clarify some of the presentation.

Page 2, lines 18-19: The author states "However, water vapour Raman lidars need to

be calibrated with an instrument measuring simultaneously for example a microwave radiometer (MWR) or radiosonde (RS) ..." This statement should be rephrased. For example ... "However, water vapour Raman lidars should be calibrated using simultaneous and collocated measurements from for example a microwave radiometer (MWR) or a radiosonde (RS).

Page 2, lines 19-20: The author states "Until now, lidars were mainly used as research instruments that did not work unattended or automatically on a routine basis." Although, the author uses the word "mainly" to qualify his statement it is still a bit misleading as there is at least one system that I'm aware of that has operated nearly continuously for over 20 years now.

Page 2, line 31: I suggest changing "In contrast to the already presented remote sensing observations..." to "By contrast, ..."

Page 2, line 32: I suggest changing "RS launches ..." to "Routine RS launches..." or "Operational RS launches..."

Page 3 lines 24-26: The author states "Calibration methods only based on RS (England et al., 1992; Mattis et al., 2002; Reichardt et al., 2012) are inappropriate for continuous monitoring of the tropospheric water vapour with Raman lidar because of their low temporal resolution." I take exception to this blanket statement. It is true that temporal resolution is an issue with the RS. However, the primary advantage in using the RS (as opposed to MWR) is that the RS provides detailed vertical information that is useful for determining height-dependent calibrations, including overlap corrections, and any possible variations in response of the detectors (eg PMTs). It would be impossible to estimate these height-dependancies using integrated water vapour measurements.

Page 3, line 34: change "the focus of the presented work is to routinely retrieve a continuous..." to something like "the focus of the present work is to develop a method that enables routine retrieval of a continuous..."

Page 4, lines 15-16: The author states "However, the overlap of both Raman channels is assumed to be identical and for that reason the overlap effect is negligible regarding water vapour measurements." This is a huge assumption that needs to be verified or justified in some way. Our system, for example, exhibits some residual overlap effect that requires correcting (and we use RS to do that).

Page 4, line 31: change "Their uncertainties amount to…" to "Their uncertainties are…"

Page 6 line 10-11: The author states "This state is then combined with the current lidar measurement yk to the filtered state xFk using the Kalman filter." The meaning here is not clear. It would make more sense if it read "This state is then combined with the current lidar measurement yk to obtain the filtered state xFk using the Kalman filter." Please clarify.

Page 7, line 2: Change "accounts" to "is equal"

Page 7, line 3: The author states "The benefit of using the logarithm is the limited range of variation…" I would think that the limited range of variation would make your job harder. Please clarify.

Page 7, line 10: change "errors at certain height levels" to "errors at different height levels"

Page 7, line 11: The author states "The second measurement vector, from now on observation vector, is given as…" I'm not clear what is meant by "from now on the observation vector". One suggestion is "The second measurement vector, which we refer to as the observation vector, is given by…." But I'm not sure if this alters your meaning. Please rephrase.

Page 7, line 17-18: The author states that the diagonal and off-diagonal elements of the MWR covariance matrix is set to 0.25K^2 and 0.01K^2, respectively. Please provide some justification (or reference) for these numbers.

Page 7, line 20: change "...amounts to..." to either "...is..." or "...was determined to be..." Which ever is more appropriate. Also please provide some justification for the uncertainty that you quote (0.1 g/kg).

Page 7, line 20-21: The author states "However, the uncertainty is increased due to the distance between the observation platform and the surface humidity sensor (see Sec. 2) and is assumed to be 0.3 g/kg." First, what is the "observation platform"? Are you referring to the lidar?. Second, if I understand correctly you are trying to account for the fact that the lidar and the surface met sensor are not collocated. Correct? The meaning is a bit obscure. Please clarify.

Page 7, line 22: Change "...certain amount..." to "...certain number..."

Page 7, line 23: The author states that 211 RS profiles were used to determine mean profiles and covariance matrix. Were these profiles taken during the span of the HOPE campaign? Please specify the time period covered by these RS profiles.

Page 7, line 24: Change "...serves as first guess..." to "...serves as a first guess..."

Page 7, line 31: Change "...cloud base due the ..." to "...cloud base due to the ..."

Page 7, line 32: Change "...using previous informations." to "...using previous information from the lidar and the RS." (If that doesn't alter your meaning)

Page 8, line 7: The author states "The transition error $e_{t;k}$ corresponds to the covariance matrix $S_{t;k}$". What do you mean by "corresponds to"? I assume this means that the diagonal elements of the covariance matrix are the (squares of the) transition errors. Please clarify the statement.

Page 8, line 11: change "...is to start with RS climatology covariance as previous covariance..." to "...is to start with the RS climatology covariance as the previous covariance..."

Page 9, line 1: insert "where" at the beginning of this line.

[Figure]

Page 9, line 2: change ". . . at time step k to the filtered state. . ." to ". . . at time step k to obtain the filtered state. . ." Page 9, line 8: change "servee" to "serve"

Page 8 and 9, equations (6) through (10): I think it is appropriate that the author present these equations in general form, as he has done. However, given the assumptions mentioned on page 8 (Hk and Mk are both unity), these equation do simplify quite a bit. I believe it would be useful to also present the simplified equations.

Page 9, line 19: The author seems to indicate that the MWR also contains a pressure sensor? If this is so then it should be mentioned in section 2.2.

Page 9, line 23: The author states "The forward modelling of the surface mixing ratio is trivial. It is a one: one translation to the lowest level of the state vector x." The second sentence is a bit confusing. What do you mean when you say "one transition to the lowest level" ? Please clarify.

Figure 5. It would be helpful to label the two boxes on the right as "lidar" and "MWR"

Page 13, line 1: change "written out Eq (13) becomes" to something like "with each term written out Eq (13) becomes"

Page 13, line 5: change ". . .matrices of the a priori and the observation. . ." to ". . .matrices of the a priori state and the observation. . ."

Page 14, line 5: change ". . .might converge at a false minimum. . ." to ". . .might converge to a false minimum. . ."

Page 14, line 17: change ". . .are no peaked. . ." to ". . .are not peaked. . ."

Page 14, line 25: change ". . .that means the higher the correlations, the smaller. . ." to ". . .the higher the correlations and the smaller. . ."

Page 15, line 7: change ". . .amount to. . ." to ". . .is. . ."

Figure 7: I'm a little perplexed by panel (d) showing the averaging kernals.Shouldnt

these functions be centered on the heights listed in the panel? Or am I missing something here.

Page 16, line 1: change "...according..." to "...corresponding..."

Page 16, line 4-5: The author states "The precise vertical information from the lidar results in small differences to the RS that is used as reference." This statement implies that the RS profile is modified by the lidar profile, and I don't believe that is the intended meaning. Also, I'm not clear what the author means by "precise." Is the author referring to the vertical resolution or low random uncertainty? Please rephrase, or eliminate this sentence entirely.

Page 16, line 14: change "...driven by the accurate a priori that..." to "...driven by the accurate a priori state that..."

Page 16 line 21: change "The according profile with..." to " The profile corresponding to..."

Page 17, line 16: change "...according..." to "...corresponding..."

Page 18, lines 3-4: The author states "In summary, the presence of a lidar measurement results in more accurate retrievals compared to RS, whereas retrievals without water vapour profiles from lidar are mainly driven by the MWR observation for example during daytime." The first part of this sentence could be interpreted to mean that the retrievals are more accurate than the RS, which is not the intended meaning. I would suggest rewording this sentence. One suggestion is: "In summary, the presence of lidar measurements results in retrievals that are in better agreement with the RS compared to retrievals without the lidar measurements. Retrievals without lidar measurements are mainly driven by the MWR observation."

Page 19, line 4: change "The according relative differences to..." to "The corresponding relative difference with..."

Page 20, line 12: Change "...according..." to "...corresponding..."

Page 20, line 14: change "...amounts..." to "...is..."

Page 20, line 13: change "breakdowns" to "malfunctions"

Page 20, line 14-15: The author states "Although, the data coverage is larger as for Raman lidar (Fig. 12a), but the vertical resolution is much coarser." This needs to be rephrased. One suggestion is "Athough the data availability for OEM_MWR is larger than the lidar, the vertical resolution is coarser."

Page 21, line 16: The author states "This issue might be caused by slightly to small determined Raman lidar calibration factors resulting in too small lidar mixing ratios." This sentence needs to be rephrased. One suggestion is "This issue might be caused by Raman lidar calibration factors that are slightly too small, resulting lidar mixing ratios that are too small."

Page 21, lines 19-20; The author states: "Hence the retrieval tends to overestimate the mixing ratio in larger heights to minimize the difference between the modelled and the observed brightness temperatures." This sentence should be reworded. One suggestion is "Hence the retrieval tends to overestimate the mixing ratio at higher altitudes in order to minimize the difference between the modelled and the observed brightness temperatures."

Figure 13: It would be helpful to add annotation indicating how the difference is defined, e.g. is it retrieval-RS or vice versa? Also in the caption the author uses the term "bias to RS." It would be clearer to say something like "mean difference between the retrieval and the RS."

Page 22 lines 11-12: The author states "... RS can have a dry bias (Miloshevich et al., 2001)." Careful here. This is a bit of an oversimplification. According to Bomin et al. (2010) the sondes used in this study (DFM-90) tend to show a daytime dry bias, but its not quite as bad the widely used Viasala sondes/

Bomin Sun, Anthony Reale,Dian J. Seidel, and Douglas C. Hunt, 2010: Comparing

radiosonde and COSMIC atmospheric profile data to quantify differences among ra-
diosonde types and the effects of imperfect collocation on comparison statistics. JGR,
115, D23104.

Figure 14: See comment for Fig13. Also, in the caption change "absorbtion" to "ab-
sorption".
* * *

---

## Author Comment (AC1) · 10 Jul 2017

The comment was uploaded in the form of a supplement:
https://www.atmos-meas-tech-discuss.net/amt-2017-77/amt-2017-77-AC1-supplement.pdf

---

## Author Comment (AC2) · 10 Jul 2017

The comment was uploaded in the form of a supplement:
https://www.atmos-meas-tech-discuss.net/amt-2017-77/amt-2017-77-AC2-supplement.pdf

---

## Author Response (AR1)

**Response to Reviewers #1 and #2**

**We like to thank the reviewers for providing helpful comments to improve the manuscript. All changes are highlighted in the manuscript file. Added text is wavy-underlined and blue, discarded text is struck out and red.**

**Response to Reviewer #1**

a) General comments:

The paper by Foth et al. presents a retrieval of water vapor from ground based microwave radiometer using an apriori computed from the previous state and collocated lidar measurements. This paper is appropriate for AMT as no science is discussed.

Before publication, the paper needs to add a discussion on how the error in the filtered profile changes as the previous state is farther in time.
**The error of the filtered profile is time independent. Every step starts with the covariance matrix of the climatology. This means that there is no relation between the previous covariance and the current covariance. The according informations are added to the text:**
**"Another possibility is to start with the RS climatology covariance ($S_{clim}$) as previous covariance matrix ($S_{k-1}$) at every consecutive time step. Using this approach the addition of the transition covariance matrix ($S_{t,k}$) can be skipped. In this application the latter approach is used which is much more stable.**
**Using Eq. (5) and the assumptions explained above, the last analysed state $\hat{x}_{k-1}$ and its covariance matrix $\hat{S}_{k-1}$ are propagated as follows:**

$$\mathbf{x}_k^{\mathrm{E}} = \mathbf{M}_k \hat{\boldsymbol{x}}_{k-1} = \hat{\boldsymbol{x}}_{k-1} \tag{6}$$

$$\mathbf{S}_k^{\mathrm{E}} = \mathbf{M}_k \hat{\mathbf{S}}_{k-1} \mathbf{M}_k^T + \mathbf{S}_{t,k} = \mathbf{S}_{\mathrm{clim}} \tag{7}$$

**where $x_k^E$ and $S_k^E$ are the estimated state and its covariance matrix, respectively. These are then combined with the lidar"**

Also, it needs to have a section/examples showing the impact of using the saturation constrain versus not using one.
**We added a small section with a figure that are showing the benefit of the supersaturation constraint.**

b) Detailed comments:

- Comments: P1L17: This sentence implies that the water vapor is some how being dictated by the IPCC, please rewrite.
  **We rewrote the sentence.**

- P1L19: Its amount is controlled mostly by the air temperature, rather by emissions. → In the stratosphere, the water vapor is controlled by the tropopause coldest temperature, do you mean in the boundary layer is controlled by sea surface temperature. Please specify and provide a reference.
  **A reference is added.**

- P1L24: About the H2O lifetime, is that in the troposphere/boundary layer? Are you sure about this number? Figure 5.23 of "Aeronomy of the Middle Atmosphere: Chemistry and Physics of the Stratosphere (Guy Brasseur and Susan Solomon)" shows a lifetime varying from hundred of years at around 20km to 1 day at around 120km.
  **This number is given by Myhre et al., 2013 in FAQ 8.1.**

- P3L6 Please describe what is the difference between this study and Han et al. 1997? Is it just that in

here it is used optimal estimation?
**We added and explanation:**
**"Han et al., (1997) presented a~method based on a KalmanFilter (Kalman,1960;Kalman and Bucy 1961) that incorporates current and past measurements followed by a statistical inversion that combines the lidar with the radiometric and climatological data."**

- P5L6 Weighting functions and jacobians are not the same. Please double check what are you showing.
  **We removed the according text passage.**

- P6L9: You are not projecting in time the last analysed state, you are just advancing this state without any modification because H is and M are the unity matrix.
  **Done as suggested.**

- Please change Figure 3: please delete the blue part of the color bar in the correlation coefficient.
  **In our opinion, the blue part of the color bar is necessary. It implies that negative correlations are possible which might happen in some cases. See for example Barrera-Verdejo et al., 2016, AMT.**

- Equation 7, why is St,k here, when you just said P8L12 than St,k can be skipped. I would assume that using RS climatology is an extreme case, if you are using the previous state from 5 min ago, the correlation and covariance matrices are going to be completely different to the ones if the previous state were from 6 hours ago (I presume that these ones will look more like the RS climatology) Please clarify.
  **Clarified. Missing information is added.**

- P10L10. In this work, it has be shown . . . This implies that in a previous section of Foth et al 2017 it has been shown that such a method is more robust, which is not the case. Change to: The cloud base of a liquid . . . (Baars, et al) which has been shown to be a more robust method for the automatic . . .
  **Done as suggested.**

- P12L6: In Rodgers there is no mention of supersaturation cost function.
  **We shifted the Rodgers reference to the sentence before.**

- Further, have you try not using such constraint. You are apriori is so tight I do not expect that it is needed.
  **We added an example showing the benefit of the constaint.**

- Equation 18 has an extra bracket.
  **No, we can't find it.**

- Figure 6 and 10: You could show the OEMMWR to showcase the impact of the lidar apriori.
  **We don't see the benefit of adding the $OEM_{MWR}$ to the time series. The benefit of the lidar as a priori is shown in more detail in the profiles in Fig. 8 and 11.**

- Figure 8 Could you add the lidar measurement to see if the bias is bigger or smaller than the OEM. Also, is this for 23:02 as in figure 7 or this are for a previous estimate.
  **Lidar measurement and a small explanation are added.**

  Further, when you do these comparisons do you apply the averaging kernels, all the MWR will look poorly if you do not include them.
  **For this comparison it is not handy to smooth the RS using the averaging kernels of the MWR to account for the limited vertical resolution of the MWR as proposed by Löhnert and Maier (2012).**

**The MWR$_{stat}$ ist just added to show that the MWR is not able to distinguish fine vertical structures. Your proposed method would be more interesting for a more precise analysis towards measurement, forward modeling and statistical representativeness of the MWR$_{stat}$ to identify a systematic bias but this would go beyond the scope of this work.**

Also MWRstat never has an error bar, please add. (also in Figure 11)
**MWR is obtained by a statistical retrieval. Statistical retrievals do not provide theoretical errors. We think adding general assumed errors of 10% could be confusing. Therefore we did not add error assumptions.**

- Figure 11: What time?
  **Time is added.**

- P19L6: For the most part, the RS profile is also within the OEM MWR uncertainty
  **Done as suggested.**

- P21L5: add shown in red after "as introduced in Fig.9"
  **Done as suggested.**

- P21L15-20: Are you implying that the lidar retrieval have a bias for altitude higher than 6km. Could you relax your apriori at those heights so that the information arises from the MWR. This needs to be fixed because in the clear sky the bias should be less than in the rest of the scenarios.
  **We changed this paragraph as follows:**
  **"Above 6 km the retrieved profiles show higher values than the RS. This bias needs to be investigated in further studies and is beyond the scope of this study."**

  Is the accuracy error shown in Figure 14 taken into account in the error characteristics shown in the previous figures.
  **The grey line shown in Fig. 14 has the same shape than the biases in the previous figures. This is an indication that the forward model causes the bias shape.**

  Also, what are the main differences between L93 and R98, do they use different absorption cross sections, if so, which ones.
  **Details are given in Rosenkranz (1998) and Liebe (1993). We added the missing information:**
  **"The differences are the line shape parameters of the 22.235 Ghz water vapour line, as well as the water vapour continuum absorption"**

  Speaking of accuracy, what is the impact of the temperature error upon your water vapor retrievals. Would it be better to use a profile derive from RS than from MWR due to the low vertical resolution of the later.
  **The impact of the uncertainty of temperature profile from the MWR is quite low. On average the MWR profile is good although it is not able to distinguish sharp structures. Using the RS profiles as input for the retrieval is not within the scope of this work which is presenting a retrieval only based on remote sensing instrument in a straightforward way.**

- P23L15: enormously is a strong word change to marginally improves.
  **Changed to 'considerably', since the RMSE is reduced by half (or even better) using the Kalman filter. This is not marginally.**

**Response to Reviewer #2**

**We like to thank the reviewer for providing helpful comments to improve the manuscript.**

a) General comments:

This paper presents an optimal estimation technique for retrievals of water vapor mixing ratio profiles by combining Raman lidar (RL) and microwave radiometer (MWR) data. The paper is well organized, and well written for the most part. I believe it represents a valuable contribution, but I found some of the text and mathematical details a bit difficult to follow. I have provided a number of comments and suggestions that I feel may help clarify some of the presentation.

b) Detailed comments:

- Page 2, lines 18-19: The author states "However, water vapour Raman lidars need to be calibrated with an instrument measuring simultaneously for example a microwave radiometer (MWR) or radiosonde (RS)..." This statement should be rephrased. For example ... "However, water vapour Raman lidars should be calibrated using simultaneous and collocated measurements from for example a microwave radiometer (MWR) or a radiosonde (RS).
  **Done as suggested.**

- Page 2, lines 19-20: The author states "Until now, lidars were mainly used as research instruments that did not work unattended or automatically on a routine basis." Although, the author uses the word "mainly" to qualify his statement it is still a bit misleading as there is at least one system that I'm aware of that has operated nearly continuously for over 20 years now.
  **'mainly' changed to 'mostly' and changed 'lidar' to 'Raman lidar'**

- Page 2, line 31: I suggest changing "In contrast to the already presented remote sensing observations..." to "By contrast, ..."
  **Done as suggested.**

- Page 2, line 32: I suggest changing "RS launches ..." to "Routine RS launches..." or "Operational RS launches..."
  **Done as suggested.**

- Page 3 lines 24-26: The author states "Calibration methods only based on RS (England et al., 1992; Mattis et al., 2002; Reichardt et al., 2012) are inappropriate for continuous monitoring of the tropospheric water vapour with Raman lidar because of their low temporal resolution." I take exception to this blanket statement. It is true that temporal resolution is an issue with the RS. However, the primary advantage in using the RS (as opposed to MWR) is that the RS provides detailed vertical information that is useful for determining height-dependent calibrations, including overlap corrections, and any possible variations in response of the detectors (eg PMTs). It would be impossible to estimate these height-dependancies using integrated water vapour measurements.
  **We changed to:**
  **"Calibration methods only based on RS (England,1992;Mattis,2002;Reichardt,2012) are often inappropriate for continuous monitoring of the tropospheric water vapour with Raman lidar because of their low temporal resolution and the requirement of regular RS launches."**

- Page 3, line 34: change "the focus of the presented work is to routinely retrieve a continuous..." to something like "the focus of the present work is to develop a method that enables routine retrieval of a continuous..."
  **Done as suggested.**

- Page 4, lines 15-16: The author states "However, the overlap of both Raman channels is assumed to be identical and for that reason the overlap effect is negligible regarding water vapour measurements." This is a huge assumption that needs to be verified or justified in some way. Our system, for example, exhibits some residual overlap effect that requires correcting (and we use RS to do that).
  **To account for the uncertainties in the overlap region we artificially increase the lidar error within the lowermost 600 m. In this range the optimal estimation modifies the profile with help of the surface measurement. However, this information was missing in the text and is now added.**

- Page 4, line 31: change "Their uncertainties amount to…" to "Their uncertainties are…"
  **Done as suggested.**

- Page 6 line 10-11: The author states "This state is then combined with the current lidar measurement y k to the filtered state xFk using the Kalman filter." The meaning here is not clear. It would make more sense if it read "This state is then combined with the current lidar measurement y k to obtain the filtered state xFk using the Kalman filter." Please clarify.
  **Done as suggested.**

- Page 7, line 2: Change "accounts" to "is equal"
  **Done as suggested.**

- Page 7, line 3: The author states "The benefit of using the logarithm is the limited range of variation…" I would think that the limited range of variation would make your job harder. Please clarify.
  **Clarified.**

- Page 7, line 10: change "errors at certain height levels" to "errors at different height levels"
  **Done as suggested.**

- Page 7, line 11: The author states "The second measurement vector, from now on observation vector, is given as…" I'm not clear what is meant by "from now on the observation vector". One suggestion is "The second measurement vector, which we refer to as the observation vector, is given by…." But I'm not sure if this alters your meaning. Please rephrase.
  **Clarified.**

- Page 7, line 17-18: The author states that the diagonal and off-diagonal elements of the MWR covariance matrix is set to 0.25K^2 and 0.01K^2, respectively. Please provide some justification (or reference) for these numbers.
  **Reference added, Barrera-Verdejo et al., 2016.**

- Page 7, line 20: change "…amounts to…" to either "…is…" or "…was determined to be…" Which ever is more appropriate. Also please provide some justification for the uncertainty that you quote (0.1 g/kg).
  **Done as suggested.**

- Page 7, line 20-21: The author states "However, the uncertainty is increased due to the distance between the observation platform and the surface humidity sensor (see Sec. 2) and is assumed to be 0.3 g/kg." First, what is the "observation platform"? Are you referring to the lidar?. Second, if I understand correctly you are trying to account for the fact that the lidar and the surface met sensor are not collocated. Correct? The meaning is a bit obscure. Please clarify.
  **Clarified.**

- Page 7, line 22: Change "…certain amount…" to "…certain number…"

**Done as suggested.**

- Page 7, line 23: The author states that 211 RS profiles were used to determine mean profiles and covariance matrix. Were these profiles taken during the span of the HOPE campaign? Please specify the time period covered by these RS profiles.
**Time period is specified.**

- Page 7, line 24: Change "...serves as first guess..." to "...serves as a first guess..."
**Done as suggested.**

- Page 7, line 31: Change "...cloud base due the ..." to "...cloud base due to the ..."
**Done as suggested.**

- Page 7, line 32: Change "...using previous informations." to "...using previous information from the lidar and the RS." (If that doesn't alter your meaning)
**No changes, besides the removed 's' behind information.**

- Page 8, line 7: The author states "The transition error e_t;k corresponds to the covariance matrix S_t;k ". What do you mean by "corresponds to"? I assume this means that the diagonal elements of the covariance matrix are the (squares of the) transition errors. Please clarify the statement.
**Clarified:**
**"The square of the transition error $\varepsilon_{t,k}$ forms the diagonal elements of the covariance matrix $S_{t,k}$."**

- Page 8, line 11: change "...is to start with RS climatology covariance as previous covariance..." to "...is to start with the RS climatology covariance as the previous covariance..."
**Done as suggested.**

- Page 9, line 1: insert "where" at the beginning of this line.
**Done as suggested.**

- Page 9, line 2: change "... at time step k to the filtered state..." to "... at time step k to obtain the filtered state..."
**Done as suggested.**

- Page 9, line 8: change "servee" to "serve"
**Done as suggested.**

- Page 8 and 9, equations (6) through (10): I think it is appropriate that the author present these equations in general form, as he has done. However, given the assumptions mentioned on page 8 (Hk and Mk are both unity), these equation do simplify quite a bit. I believe it would be useful to also present the simplified equations.
**Simplification is not trivial in these cases since H isn't a quadratic matrix. Omitting H is not possible/allow due to the fact that the matrix multiplication is dependent on the matrix dimension. Simplification is done for M and for the covariance matrix of the climatology.**

- Page 9, line 19: The author seems to indicate that the MWR also contains a pressure sensor? If this is so then it should be mentioned in section 2.2.
**Missing paragraph is added to the MWR section:**
**"The MWR was also equipped with a standard meteorological weather station measuring temperature, pressure and relative humidity. These values are only used to calculate the pressure profile that is used in the forward model. The surface values needed for the optimal estimation originate in the surface tower measurement which is much more accurate. Arising pressure uncertainties result in negligible deviation in the modelled brightness temperatures."**

- Page 9, line 23: The author states "The forward modeling of the surface mixing ratio is trivial. It is a one: one translation to the lowest level of the state vector x." The second sentence is a bit confusing. What do you mean when you say "one transition to the lowest level" ? Please clarify.
  **We changed the ':' to 'to'. It means one to one. We think it is now more clear.**

- Figure 5. It would be helpful to label the two boxes on the right as "lidar" and "MWR"
  **We were also considering that, however it isn't the lidar and MWR. Actually, the observation vector comprises both MWR and surface measurement and the a priori isn't the lidar but the Kalman filtered lidar profile.**

- Page 13, line 1: change "written out Eq (13) becomes" to something like "with each term written out Eq (13) becomes"
  **Done as suggested.**

- Page 13, line 5: change "...matrices of the a priori and the observation..." to "...matrices of the a priori state and the observation..."
  **Done as suggested.**

- Page 14, line 5: change "...might converge at a false minimum..." to "...might converge to a false minimum..."
  **Done as suggested.**

- Page 14, line 17: change "...are no peaked..." to "...are not peaked..."
  **Done as suggested.**

- Page 14, line 25: change "...that means the higher the correlations, the smaller..." to "...the higher the correlations and the smaller..."
  **Done as suggested.**

- Page 15, line 7: change "...amount to..." to "...is..."
  **Done as suggested.**

- Figure 7: I'm a little perplexed by panel (d) showing the averaging kernels. Shouldn't these functions be centered on the heights listed in the panel? Or am I missing something here.
  **In principle you are right. But as written in the text this is only the case for an observation that is highly sensitive with a high amount of vertical information. Unfortunately the vertical information content of water vapour from MWR is limited (degree of freedom around 2) resulting in non-peaked smooth curves.**

- Page 16, line 1: change "...according..." to "...corresponding..."
  **Done as suggested.**

- Page 16, line 4-5: The author states "The precise vertical information from the lidar results in small differences to the RS that is used as reference." This statement implies that the RS profile is modified by the lidar profile, and I don't believe that is the intended meaning. Also, I'm not clear what the author means by "precise." Is the author referring to the vertical resolution or low random uncertainty? Please rephrase, or eliminate this sentence entirely.
  **Rephrased as follows:**
  **"The ability of the lidar to perform precise water vapour measurements results in small differences to the reference RS."**

- Page 16, line 14: change "...driven by the accurate a priori that..." to "...driven by the accurate a

priori state that..."
**Done as suggested.**

- Page 16 line 21: change "The according profile with..." to " The profile corresponding to..."
**Done as suggested.**

- Page 17, line 16: change "...according..." to "...corresponding..."
**Done as suggested.**

- Page 18, lines 3-4: The author states "In summary, the presence of a lidar measurement results in more accurate retrievals compared to RS, whereas retrievals without water vapour profiles from lidar are mainly driven by the MWR observation for example during daytime." The first part of this sentence could be interpreted to mean that the retrievals are more accurate than the RS, which is not the intended meaning. I would suggest rewording this sentence. One suggestion is: "In summary, the presence of lidar measurements results in retrievals that are in better agreement with the RS compared to retrievals without the lidar measurements. Retrievals without lidar measurements are mainly driven by the MWR observation."
**Rephrased.**

- Page 19, line 4: change "The according relative differences to..." to "The corresponding relative difference with..."
**Done as suggested.**

- Page 20, line 12: Change "...according..." to "...corresponding..."
**Done as suggested.**

- Page 20, line 14: change "...amounts..." to "...is..."
**Done as suggested.**

- Page 20, line 13: change "breakdowns" to "malfunctions"
**Done as suggested.**

- Page 20, line 14-15: The author states "Although, the data coverage is larger as for Raman lidar (Fig. 12a), but the vertical resolution is much coarser." This needs to be rephrased. One suggestion is "Athough the data availability for OEM_MWR is larger than the lidar, the vertical resolution is coarser."
**Done as suggested.**

- Page 21, line 16: The author states "This issue might be caused by slightly to small determined Raman lidar calibration factors resulting in too small lidar mixing ratios." This sentence needs to be rephrased. One suggestion is "This issue might be caused by Raman lidar calibration factors that are slightly too small, resulting lidar mixing ratios that are too small."
**Done as suggested.**

- Page 21, lines 19-20; The author states: "Hence the retrieval tends to overestimate the mixing ratio in larger heights to minimize the difference between the modelled and the observed brightness temperatures." This sentence should be reworded. One suggestion is "Hence the retrieval tends to overestimate the mixing ratio at higher altitudes in order to minimize the difference between the modelled and the observed brightness temperatures."
**Done as suggested.**

- Figure 13: It would be helpful to add annotation indicating how the difference is defined, e.g. is it retrieval-RS or vice versa? Also in the caption the author uses the term "bias to RS." It would be

clearer to say something like "mean difference between the retrieval and the RS."
**Done as suggested.**

- Page 22 lines 11-12: The author states "... RS can have a dry bias (Miloshevich et al., 2001)." Careful here. This is a bit of an oversimplification. According to Bomin et al. (2010) the sondes used in this study (DFM-90) tend to show a daytime dry bias, but its not quite as bad the widely used Viasala sondes/Bomin Sun, Anthony Reale,Dian J. Seidel, and Douglas C. Hunt, 2010: Comparing radiosonde and COSMIC atmospheric profile data to quantify differences among radiosonde types and the effects of imperfect collocation on comparison statistics. JGR, 115, D23104.
  **Removed.**

- Figure 14: See comment for Fig13. Also, in the caption change "absorbtion" to "absorption".
  **Done as suggested.**

[revised manuscript text omitted]